# Trophic Diversity of a Fish Community Associated with a *Caulerpa prolifera* (Forsskål) Meadow in a Shallow Semi-Enclosed Embayment

Maria Maidanou [1,2], Panayota Koulouri [1,*], Paraskevi K. Karachle [3], Christos Arvanitidis [1], Drosos Koutsoubas [2] and Costas Dounas [1]

[1] Institute of Marine Biology, Biotechnology and Aquaculture, Hellenic Centre for Marine Research, Gournes Pediados, P.O. Box 2214, 71003 Heraklion, Greece; mariam@hcmr.gr (M.M.); arvanitidis@hcmr.gr (C.A.); kdounas@hcmr.gr (C.D.)

[2] Faculty of Environment, Department of Marine Sciences, University of the Aegean, 81100 Mytilene, Greece; drosos@aegean.gr

[3] Institute of Marine Biological Resources and Inland Waters, Hellenic Centre for Marine Research, Athens-Sounio Ave, P.O. Box 712, 19013 Anavyssos, Greece; pkarachle@hcmr.gr

* Correspondence: yol72@hcmr.gr; Tel.: +30-2810-337716

**Abstract:** This study investigates the trophic diversity of fishes living in a meadow of *Caulerpa prolifera* on a bimonthly basis between May 2006 and April 2007 in a semi-enclosed coastal marine ecosystem of the Mediterranean Sea (Elounda Bay, Crete Island). The study area is shallow and protected from waves, and it is covered by a *C. prolifera* bed, characterized by high organic input and a highly diverse macrobenthic community. Feeding patterns of the fish, investigated on the basis of stomach content analyses, were described in terms of numerical abundance and frequency of occurrence of prey taxa. A total of 1642 fish individuals, belonging to 17 species, were examined. In total, 45,674 prey individuals were identified belonging to 110 prey taxa, most of which were Malacostraca including their larvae and Copepoda (41,175 individuals identified to 71 taxa). Four different trophic groups were identified: herbivorous, pelagic, benthic (hyperbenthic) and piscivorous. Trophic diversity patterns of the fish species studied were also compared to the relative availability of macrobenthic and zooplanktonic taxa during the same period in the study area. The coexistence of many different, mostly benthic but also pelagic, fishes and their juveniles implies their high trophic flexibility, which is probably important for their survival in this particular habitat. Results of the present study provide basic knowledge on trophic diversity and interactions in the marine ecosystem and, therefore, some evidence as to the protection value of this particular habitat, which is essential for the implementation of a multispecies approach to decision-makers and managers of fisheries sources of the region.

**Keywords:** marine demersal fish; fish juveniles; nursery ground; feeding patterns; trophic diversity; stomach content analysis; *Caulerpa prolifera*; prey selectivity; prey availability; eastern Mediterranean

## 1. Introduction

Though coastal marine habitats are very productive and diversified ecosystems, they are under continuous pressure and threat due to human activities [1]. Most of these habitats provide high food resource availability and protection against predators for the inhabiting biota, thus supporting highly diversified and abundant populations of invertebrates and fish throughout their entire life history [2]. Because of their high diversity and productivity, these nearshore marine ecosystems are considered to be "nursery grounds" [2] or "effective juvenile habitats" [3]; therefore, their protection, management and conservation are considered to be of high importance.

Feeding ecology of marine fishes utilizing specific habitats highlight their role in ecological interactions, community structure and function of marine ecosystems [4–6]. Despite its value and the progress achieved through the development of molecular techniques and

modelling tools, the traditional approach, i.e., taxonomic identification of prey taxa, for studying this field is still scanty. However, it comprises basic knowledge and, therefore, it is essential in order to have a comprehensive view leading to holistic management approaches and conservation strategies [7].

Many studies on feeding ecology have been conducted for demersal fish assemblages e.g., [4,8,9]. However, far fewer refer to the trophic structure of fishes inhabiting vegetated systems (e.g., seagrass meadows), which are considered to play a fundamental role in maintaining populations of commercially exploited fish and invertebrate species by providing nursery areas for the successful development of juveniles, feeding areas for different life-history stages and refuges from predation, e.g., [10–13]. Moreover, research on fish feeding habits or preferences in relation to their prey availability are relatively limited, e.g., [14–18]. In particular, marine macroalgal-dominated habitats have so far received very little attention [19,20]. Furthermore, there are just a few studies on trophic structure of fish species associated with habitats invaded by non-indigenous macroalgal species as this knowledge is important to track their impacts [21–24].

The present study investigates the trophic diversity of the fish species associated with a *Caulerpa prolifera* (Forsskål) bed in the semi-enclosed Elounda Bay of the Cretan Sea (eastern Mediterranean). Although *C. prolifera* beds have been recorded in many coastal areas of the Mediterranean Sea, there are just a few studies dealing with the associated macrofaunal and fish assemblages [19,20]. These assemblages seem to be comparable, in terms of species richness, to those of seagrass meadows, verifying the hypothesis that the physical structure per se is one of the main factors affecting them. Moreover, studies on the feeding habits of fish referring to the marine environment of the island of Crete are scarce [4,18,25]. The study area is characterized by high organic input and hosts a highly diversified macrobenthic community [20]. This shallow embayment has proven to be an important nursery ground for fishes, many of which represent main target species for commercial fisheries, thus contributing to the conservation and maintenance of marine biological resources of the wider area [19,26]. The main objectives of the study were (a) to explore the feeding patterns of the fish species, mostly benthic but also pelagic, and their juveniles in the study area and identify different trophic groups; (b) to investigate differences in the diet of each fish species in relation to body size and temporal occasion; and (c) to examine prey selectivity of fishes taking into account the macrobenthic and zooplanktonic taxa availability in the study area.

## 2. Materials and Methods

### 2.1. Study Area

Elounda Bay (total area: 6.5 km²) is a shallow, semi-enclosed coastal marine ecosystem relatively isolated from the outer area of Mirabello Bay (Figure 1). The present study was carried out in the inner muddy shallow part of the Bay, characterized by the presence of a continuous *C. prolifera* meadow and covering an area of 4.7 km², and depths ranging between 2 and 9 m (Figure 1). The environmental variables in the water column and the sediment of the study area were described in detail in [20]. However, it should be noted that the study area is characterized by strong seasonality, with sea water temperatures ranging from 13 °C in winter to 25 °C in summer. Nutrients and organic matter concentrations indicate an oligotrophic to mesotrophic marine ecosystem without any severe impacts, despite the tourist activities taking place in the area [27].

### 2.2. Sampling Design and Techniques

Fish samples were collected with a boat seine during daylight. Each haul swept an area of approximately 0.006 km². A single haul was taken in each of the following months: May 2006, July 2006, September 2006, November 2006, February 2007 and April 2007 for a total of six sampling hauls at one sampling site (Figure 1). The seine net was used (cod-end mesh size: 8 mm bar length), operated from a local fishing boat to sample the fishes. All the captured fish were identified to species level (Table 1), and a subsample



of thirty specimens of each species per haul (where possible) was randomly selected for stomach content analysis. The sample material was then fixed in 10% formalin on board ship and transferred to the laboratory for further analysis.

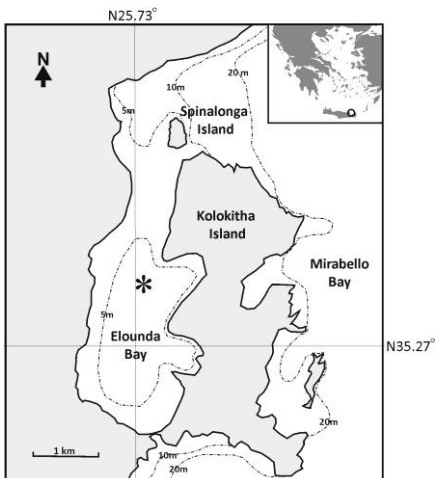

**Figure 1.** Map showing the study area of Elounda Bay. Asterisk represents the sampling site. Depth contours of 5 m, 10 m and 20 m are also indicated.

**Table 1.** Fish species and number of specimens examined for stomach content analysis during all six sampling occasions in the study area. The numbers of empty stomachs, in terms of vacuity index, are also shown. Size range of the total length (TL) of the fish specimens is given. $L_m$: length at first maturity for each species is also included.

| Family | Fish Species | Size Range of TL (mm) | No Individuals | $L_m$ (mm) | Juveniles (%) | Vacuity Index (%) | Number of Prey Taxa |
|---|---|---|---|---|---|---|---|
| Atherinidae | *Atherina boyeri* Risso, 1810 | 52.55–90.25 | 119 | 58 | 5.9 | 5.00 | 11 |
| Blenniidae | *Parablennius tentacularis* (Brünnich, 1768) | 38.95–97.44 | 126 | 75–134 | 24–82 | 12.70 | 36 |
| Centracanthidae | *Spicara maena* (Linnaeus, 1758) | 82.47–108.23 | 61 | 103 | 98.4 | 6.56 | 14 |
| Centracanthidae | *Spicara smaris* (Linnaeus, 1758) | 48.88–114.27 | 180 | 91 | 92.8 | 1.11 | 13 |
| Gobiidae | *Gobius niger* Linnaeus, 1758 | 34.72–125.36 | 177 | 54–80 | 61–100 | 2.26 | 56 |
| Labridae | *Symphodus cinereus* (Bonnaterre, 1788) | 45.80–100.70 | 102 | 40 | 0.0 | 3.92 | 35 |
| Monacanthidae | *Stephanolepis diaspros* Fraser-Brunner, 1940 | 31.72–51.48 | 2 | 80–106 | 100.0 | 0 | 2 |
| Mullidae | *Mullus barbatus* Linnaeus, 1758 | 51.70–173.35 | 150 | 111 | 94.0 | 1.33 | 69 |
| Mullidae | *Mullus surmuletus* Linnaeus, 1758 | 54.12–185.90 | 41 | 161 | 97.6 | 0 | 36 |
| Serranidae | *Serranus hepatus* (Linnaeus, 1758) | 30.01–90.72 | 167 | 78 | 97.6 | 8.98 | 51 |
| Siganidae | *Siganus luridus* (Rüppell, 1829) | 31.03–112.06 | 100 | 142 | 100.0 | 31.00 | 1 |
| Sparidae | *Boops boops* (Linnaeus, 1758) | 53.55–162.07 | 152 | 143 | 99.3 | 3.95 | 23 |
| Sparidae | *Diplodus annularis* (Linnaeus, 1758) | 35.90–99.53 | 46 | 112 | 100.0 | 2.17 | 26 |
| Sparidae | *Pagellus acarne* (Risso, 1827) | 44.71–93.73 | 126 | 160 | 100.0 | 4.76 | 23 |
| Sparidae | *Pagrus* (Linnaeus, 1758) | 39.10–162.08 | 46 | 266 | 100.0 | 6.52 | 23 |
| Sphyraenidae | *Sphyraena sphyraena* (Linnaeus, 1758) | 85.40–245.24 | 46 | 230–260 | 100.0 | 13.04 | 2 |
| Tetraodontidae | *Lagocephalus sceleratus* (Gmelin, 1789) | 111.38 | 1 | 433 | 100.0 | 0 | 2 |
| | **Total** | | **1642** | | | **0–31** | **110** |

After sampling, all individuals were measured to the nearest mm (total length, TL) and weighed to the nearest 0.01 g in the laboratory. Juveniles were defined on the basis of individual body size ($L_m$: length at first maturity) for each species, which was taken into account according to previous work [28–30]. Size range of $L_m$ was included only for *Gobius niger*, *Parablennius tentacularis*, *Sphyraena sphyraena* and *Stephanolespis diaspros* according to

references from different areas of the Mediterranean Sea or due to different sizes of sexes. The stomach and intestine, or in the case of labrid species (which generally have a poorly defined stomach) the anterior half of the alimentary tract, were then dissected and wet-weighed. The organisms found as prey in the fish stomachs were initially identified to major taxonomic prey categories (Appendix A) and also counted. In particular, macrobenthic Polychaeta, Mollusca and Crustacea were subsequently identified to species level, where possible, and also counted (Appendix B).

Data of macrobenthos and zooplankton collected during the same sampling occasions in the study area are included in order to determine the availability of prey and thus to be able to determine the selectivity of the fishes under examination. Three tows of approximately 30 m length were performed along the study area during each sampling occasion using an epibenthic sledge (0.5 mm mesh size) in order to sample macrofauna. Sampling design and data on macrobenthos are given in detail in [20]. Five vertical hauls were also taken from ~1 m above the seabed to the water surface on each sampling occasion using a plankton net (0.5 mm mesh size) in order to collect zooplankton [27]. The material collected was immediately fixed with 10% formalin and sorted under a dissecting microscope upon return to the laboratory.

### 2.3. Data Analysis

The following three indices were used for the estimation of the contribution of each prey item to the diet of each fish species and characterization of the trophic group: (a) percentage numerical abundance (% N), i.e., the number of each prey taxon in all stomachs (non-empty) expressed as a percentage of the total number of prey taxa in all stomachs; (b) percentage frequency of occurrence (% F), i.e., the number of stomachs in which a prey taxon was found as a percentage of the total number of stomachs (non-empty); (c) the vacuity index (% VI), i.e., the number of empty and nearly empty stomachs as a percentage of the total number of stomachs analyzed [31–33]. On the basis of % N contribution of each prey, the fractional trophic level (TROPH) of the species was estimated, using the routine for qualitative data of TrophLab (ICLARM: Manila, Philippines) [34]. TROPH is estimated as the contribution of the TROPH values of each prey (TROPHj) in the diet, increased by one (1) and is calculated according to the formula:

$$TROPH_i = 1 + \sum_{j=1}^{G} DC_{ij} \times TROPH_j$$

where DCij is the % N of prey (i) in the diet of consumer (i).

A cluster analysis (using group average linkage) was performed using the Bray–Curtis similarity coefficient [35] based on numerical abundance matrices of prey taxa (species where possible) found in the stomachs of the fish species examined in order to identify different trophic groups. The data were transformed to log (x + 1) prior to analysis. For the detection of significant differences between diets of fish species, the one-way analysis of similarity test (ANOSIM) was applied [36]. The similarity percentage (SIMPER) procedure was applied for the investigation of prey taxa contribution to the similarity of the above-mentioned groups. Because of the low number of individuals, the stomach contents of fish species *L. sceleratus* (1 specimen) and *Stephanolepis diaspros* (2 specimens) were excluded from the multivariate analysis. *Siganus luridus* and *Sphyraena sphyraena* were also excluded. *Siganus luridus* was excluded as its prey, i.e., fragments of *C. prolifera*, cannot be expressed by relative abundance and frequency of occurrence. *Shyraena sphyraena* was also excluded as all individuals were found only during one sampling occasion (July 2006, except for one specimen caught in November 2006) feeding almost exclusively on fishes. The PRIMER v6 (Plymouth, UK) statistical software package was used for the above-mentioned data analyses. In order to investigate prey selectivity of the fish species, percentage numerical abundance (%N) of the taxa collected in the study area with the sledge and the plankton

net during the same sampling occasions was compared with the percentage numerical abundance (%N) of the prey taxa found in the fish stomachs.

## 3. Results

A total of 1642 individual fishes, belonging to 17 species, were examined (Table 1). A total of 45,674 prey individuals were identified to 110 prey taxa (Appendix B). The highest diversity of prey taxa (69) was observed for *Mullus barbatus*, while the highest number of prey individuals (11,666) was found in the stomachs of *S. smaris* (Table 1). The dietary composition of the fish species consisted mostly of crustacean taxa as 41,175 individuals were identified to 71 taxa of Malacostraca (Appendix B).

The similarity dendrogram, based on the numerical abundance (%N) matrices of the prey taxa in the stomachs of the fish species, comprised three different trophic groups based on the types of food examined (Figure 2): (i) Pelagic trophic group included species *Boops boops*, *S. smaris*, *S. maena*, *Atherina boyeri*, *Diplodus annularis* and *P. acarne*. The diet of these fish species was characterized by planktonic copepods according to the results of SIMPER analysis (Table 2). (ii) Benthic trophic group I included the species *M. barbatus*, *M. surmuletus*, *Gobius niger*, *Parablennius tentacularis*, *Symphodus cinereus* and *Serranus hepatus*. The diet of most of these fish species consisted of a broad range of prey taxa (Table 2). However, *M. barbatus* and *G. niger* seem to have a very specific diet. (iii) The diet of species *Pagrus pagrus* comprised benthic trophic group II based on decapods. An ANOSIM test showed differences which were statistically significant between the three trophic groups (R = 0.52, *p* < 0.001).

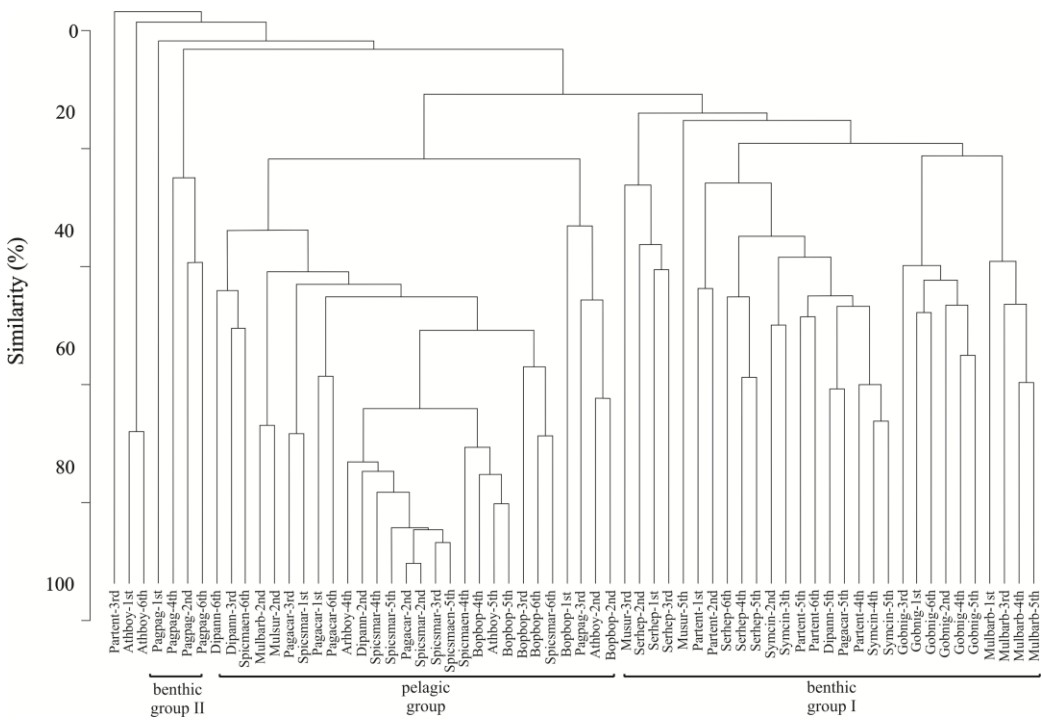

**Figure 2.** Similarity dendrogram based on the relative abundance of prey taxa matrix of the common fish species examined during the six sampling occasions in the study area.

**Table 2.** Results of SIMPER analysis.

| Taxa | Averaged Numerical Abundance | Averaged Similarity | Contribution % |
|---|---|---|---|
| Group 1 Average similarity: 41.59 | | | |
| Copepoda | 59.37 | 38.15 | 91.71 |
| Group 2 Average similarity: 18.46 | | | |
| Copepoda | 16.15 | 4.00 | 21.69 |
| Larvae insects | 13.72 | 2.26 | 12.27 |
| *Aora spinicornis* | 11.20 | 2.17 | 11.74 |
| *Leptochelia* sp. | 4.67 | 1.55 | 8.38 |
| *Caprella acanthifera discrepans* | 3.09 | 1.12 | 6.04 |
| *Microdeutopus stationis* | 3.20 | 1.05 | 5.69 |
| *Cymodoce truncata* | 5.20 | 1.00 | 5.42 |
| Varia | 3.32 | 0.91 | 4.92 |
| *Microdeutopus versiculatus* | 1.31 | 0.38 | 2.04 |
| *Caprella acanthifera* | 2.61 | 0.38 | 2.03 |
| *Phtisica marina* | 2.77 | 0.36 | 1.96 |
| Fish | 1.86 | 0.36 | 1.93 |
| *Paguristes syrtensis* | 1.83 | 0.33 | 1.81 |
| *Abra alba* | 1.83 | 0.30 | 1.64 |
| *Ericthonius* sp. | 1.15 | 0.29 | 1.56 |
| *Caprella rapax* | 1.85 | 0.23 | 1.22 |
| Group 3 Average similarity: 12.52 | | | |
| Varia | 26.17 | 6.32 | 50.46 |
| *Paguristes syrtensis* | 17.03 | 3.78 | 30.16 |
| *Liocarcinus navigator* | 7.64 | 1.82 | 14.52 |

The composition of diet of the fish species studied, in terms of numerical abundance and frequency of occurrence of the major taxonomic prey categories, is shown in Appendix A. In the same Appendix the fractional trophic levels (TROPH) of the species, as estimated in the present study, are given with the corresponding values from FishBase [28] and those from the Mediterranean Sea [4]. Crustacean larvae were the dominant prey taxon (85% N and 97% F) for the small individuals of *B. boops* caught in July (Appendix A) despite their relatively low availability in the study area (Figure 3). On the contrary, copepods were its dominant prey (maximum values of 79.4–90.3% N and 92.3–96.6% F) when larger individuals of this species were caught (Appendix A). Individuals of *S. smaris* fed almost exclusively on copepods (Appendix A), the most abundant planktonic animal group in the study area (Figure 3). Copepods and crustacean larvae were also among the dominant components of the diet of individuals of *A. boyeri* and *S. maena* (Appendix A). Small individuals of the fish species *P. acarne* (Appendix A) preyed almost exclusively on copepods (78.2–99.5% N and 100% F). The few larger ones caught in February seemed mostly to prefer amphipods and tanaids (Appendices A and B).

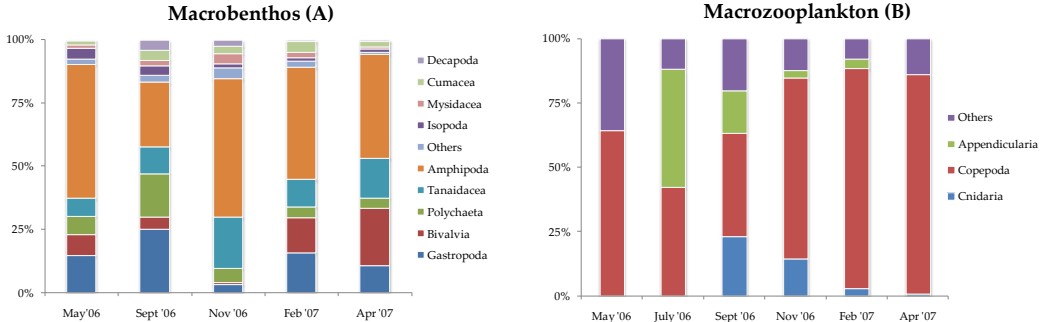

**Figure 3.** Relative abundance (% N) of the most important macrobenthic (**A**) and zooplanktonic (**B**) groups collected during the six sampling occasions in the study area.

The fish species *M. barbatus*, *G. niger*, *P. tentacularis*, *S. hepatus*, *S. cinereus* and *M. surmuletus* fed on a broad variety of dietary items; however, each species was characterized by its specific prey taxa (Appendices A and B). The smallest individuals of *M. barbatus* preyed mostly on copepods (41.5–91.2% N and 72.4–86.2% F). The common prey items for larger individuals of this species caught in May and September were polychaetes, copepods, bivalves and amphipods. The dominant prey for *G. niger* was insect larvae (35.6–78.8% N and 43.3–88.9% F, Appendix A).

The bulk of the diet of *P. tentacularis*, *S. hepatus*, *S. cinereus* and *M. surmuletus* was made up mostly of malacostracans (Appendices A and B). In particular, amphipods made the largest contribution to the diet of *P. tentacularis* (maximum values of 83.8–85.9% N and 81.8–100% F), especially in November when the smallest individuals were caught (Appendices A and B). Amphipods also made a large contribution to the diet of small individuals of *S. hepatus* (maximum values of 71.5–82.9% N and 81.5–100% F, Appendices A and B). The individuals of *S. cinereus* fed mostly on amphipods and often on tanaids (31.9–83.8% N and 82.1–100% F, 10.1–29.7% N and 60.7–81.8% F, respectively, Appendices A and B). The small individuals of *M. surmuletus*, which were caught especially in July (Appendix A), mostly fed on copepods (88.8% N and 46.7% F). Larger specimens of this species caught in September and February seemed to prefer decapods followed by amphipods (Appendices A and B).

As far as *S. luridus* is concerned, fragments of *C. prolifera* dominated its diet, except for a few amphipods. Amphipods, tanaids and copepods were common prey items for the small individuals of *D. annularis* found in the study area, while decapods were common in the stomachs of the small individuals of *P. pagrus* (Appendices A and B). Stomach contents of the small individuals of the species *S. sphyraena* examined in July were almost exclusively dominated by other fishes (98% N and 97% F). Finally, three small individuals of the non-indigenous species *L. sceleratus* (TL = 111.38 mm) and *S. diaspros* (31.72 mm and 51.48 mm) were found to feed on small crustaceans (e.g., *Leptochelia* sp. and *Microdeutopus* sp.).

## 4. Discussion

The present study attempts to shed some light on the trophic diversity of mostly benthic but also pelagic fish species associated with a *C. prolifera* bed in a coastal marine ecosystem of the Mediterranean Sea. The presence of a *C. prolifera* meadow in Elounda Bay has led to the settlement of a highly diversified macrobenthic faunal community [20] and relatively abundant zooplanktonic populations [27]. All these faunal organisms support an important feeding ground for several fish species and their juveniles living within this particular habitat. Two main strategically different trophic groups can be distinguished in the study area: a) fishes that had a relatively narrow food spectrum concerning mainly planktonic copepods, and b) fish species that prefer feeding on a highly diverse dietary composition consisting mostly of benthic and especially hyperbenthic small crustaceans (e.g., amphipods, isopods, tanaids), apart from *Pagrus pagrus* which seems to prey almost exclusively on decapods [37]. *Siganus luridus*, regardless of its size, feeds on fragments of algae [38,39] such as *C. prolifera* in the study area. The few amphipods found in the stomachs should be considered as accidental prey, having been consumed along with the algae [39]. Meanwhile, juveniles of the piscivorous predator *S. sphyraena* [40,41] occurred only sporadically in the study area. Temporal variations of the diet were also observed in most of the fish species most probably due to the important factors of fish body size and prey availability in the marine environment [15,17].

In general, the fractional trophic levels of the species studied fell within the range of previously reported ones [4]. Only in the case of the two *Spicara* commercial species the values estimated here were lower than those previously reported, most probably due to a strong preference towards highly available Copepoda and confirming their pelagic trophic behavior [18,42–44]. This preference further supports the hypothesis on the importance of these two species on the flow of energy in the food web [45]. A copepod-based diet was also characteristic of commercial species *A. boyeri* and *B. boops* [46–48], as prey abundance is considered to be one of the major factors influencing the choice of prey [14,17]. However, in

July, their diet was dominated by crustacean larvae, indicating food selection that often occurs for available prey but of low abundance in the marine environment [49,50]. Though *P. acarne* and *D. annularis* are not considered to be pelagic feeders [18,42,44,51,52], their juveniles found in the study area fed on planktonic copepods probably as an effect of body size [15].

The residents *G. niger* and *P. tentacularis* of Elounda Bay appeared to be opportunistic in their trophic behavior as they consume a wide food spectrum [18,29,42,53]. *Mullus barbatus* is also characterized by opportunistic feeding habits, while *M. surmuletus* seems to have a more specialized feeding behavior [18,53–58]. However, most of the individuals of these two commercial species examined in Elounda Bay were juveniles, especially in July, and the most frequent and abundant prey items in their stomachs were planktonic copepods and small hyperbenthic crustaceans. This corresponds to a study in north Aegean Sea where fishes in their early life history tend to a planktonic-oriented diet, regardless of their feeding habits as adults [59]. The feeding behavior of *S. cinereus* and *S. hepatus* also indicated a degree of opportunism exploiting the most commonly available food resources [14,17]. The few individuals of *P. pagrus* seem to select decapods [37] as they comprise available prey but were of low abundance in the study area.

The juveniles of *M. barbatus*, *M. surmuletus*, *P. acarne*, *D. annularis* and the pelagic species *A. boyeri*, *B. boops*, *S. smaris* and *S. maena* mainly consumed the same food resources, i.e., the highly abundant planktonic copepods, which are also rich in highly unsaturated fatty acids (HUFAs), thus more energetically profitable [60]. However, the taxonomic identification of copepods to species level could reveal a better use of the food resources in the marine environment for avoiding competition. Nevertheless, the extremely high abundance of planktonic copepods precludes the assumption for food competition in the present case, which is likely only if food resources are scarce [61]. Furthermore, the different relative importance of prey taxa (numerical abundance and frequency of occurrence) and the low similarity of multivariate analysis of the numerical abundance data suggest a considerable sharing of food resources, which seems to limit inter- and intra-specific competition in Elounda bay [62]. Though more than 300 macrofaunal taxa were identified in the study area [20,27], only 73 were used as food items by the fish species examined (Appendix B). Their principal prey consisted mostly of small hyperbenthic crustaceans (e.g., amphipods such as *A. spinicornis*, *Caprella* spp., *P. marina*, *M. stationis*, *Ericthonius* sp. the tanaid *Leptochelia* sp. and the isopod *C. truncata*), confirming their role as an important link between benthos and fish [63]. The optimal foraging theory suggests that, apart from size and relative abundance, prey characteristics such as distribution, accessibility and mobility, energy content and handling time determine prey profitability [64]. The preference for hyperbenthic crustaceans in the study area can be attributed to their availability, which is also a function of their behavior and distribution [18,63]. Hyperbenthic crustaceans are accessible to predators as they are active either at the sediment surface or a few centimeters above the sea floor and thus can easily be encountered, caught and preyed upon. Finally, crustaceans are, in general, an important source of prey due to their high quality of calorific energy for predators [65].

In conclusion, the coexistence of the different fish species and their juveniles in Elounda Bay implies their high trophic flexibility and their ability to partition available food resources. However, the possible interconnection between prey and predators reflects a certain degree of opportunistic feeding, which is probably fundamental for their survival in this particular habitat. Results of the present study concerning these fish species and their trophic structure can be used as a proxy in other areas of the Mediterranean Sea of similar diversity covered either by seagrass or macroalgal meadows, which also comprise nursery grounds and refuges from predators. Moreover, Elounda Bay is a particular marine environment threatened by tourist activities, as large boats transfer tourists to and from Spinalonga island (see Figure 1) on a daily basis during the summer period, causing resuspension of the sediments that most probably have a negative effect on the conservation of the *C. prolifera* bed and therefore on maintenance of its marine biological resources. Flood events also take place in the study area occasionally. Therefore, basic knowledge (e.g.,

area cover of *C. prolifera* canopy, macrofaunal and fish assemblages, trophic structure) is essential in order to regularly monitor, protect and conserve this valuable marine ecosystem as well as further achieve implementation of a multispecies approach for decision-makers and managers of fisheries sources of the region.

**Author Contributions:** Conceptualization, C.D. and P.K.; methodology, C.D., P.K. and P.K.K.; software, M.M., P.K. and P.K.K.; validation, M.M., P.K., P.K.K., C.A. and D.K.; formal analysis, M.M., P.K. and P.K.K.; investigation, M.M., P.K., P.K.K., C.A. and D.K.; resources, C.D.; data curation, M.M., P.K., P.K.K.; writing—original draft preparation, M.M., P.K. and P.K.K.; writing—review and editing, P.K., P.K.K. and C.D.; visualization, M.M., P.K. and P.K.K.; supervision, C.D., P.K. and D.K.; project administration, C.D.; funding acquisition, C.D. and P.K. All authors have read and agreed to the published version of the manuscript.

**Funding:** This research was funded by the former Lasithi Prefecture of Crete Island (Greece) within the framework of the project: Environmental Study of Elounda Bay.

**Institutional Review Board Statement:** Not applicable.

**Informed Consent Statement:** Not applicable.

**Data Availability Statement:** Not applicable.

**Acknowledgments:** The authors are grateful for the critical reading of the manuscript made by A. Eleftheriou as well as the comments on the manuscript made by M. Eleftheriou.

**Conflicts of Interest:** The authors declare no conflict of interest.

## Appendix A

Numerical abundance (%N), frequency of occurrence (%F) and number of species of the main taxonomic prey categories found as prey in the stomachs of the fish species examined during the different occasions in the study area. Number of individuals with empty stomachs is shown in parentheses. Mean total length (TL) in mm and mean wet biomass in g are also shown per individual during each sampling occasion in the study area.

| Taxonomic Prey Categories | May '06 | | July '06 | | Sept '06 | | Nov '06 | | Feb '07 | | Apr '07 | |
|---|---|---|---|---|---|---|---|---|---|---|---|---|
| | %N | %F | %N | %F | %N | %F | %N | %F | %N | %F | %N | %F |
| *Atherina boyeri* Individuals/Juveniles | 5 | 0 | 42 (6) | 1 | | | 30 | 4 | 30 | 2 | 11 | 0 |
| Mean TL (mm)/mean wet weight (g) | | 64.85/1.6 | | 64.21/1.7 | | | | 64.60/0.2 | | 64.64/1.7 | | 67.00/1.8 |
| Polychaeta | | | 0.9 | 10.5 | | | | | 0.1 | 6.7 | | |
| Crustacea (larvae) | 11.8 | 20.0 | 52.9 | 68.4 | | | 1.6 | 6.9 | 0.3 | 16.7 | | |
| Copepoda | | | 18.9 | 10.5 | | | 96.4 | 96.6 | 93.4 | 100.0 | | |
| Ostracoda | | | | | | | | | 0.1 | 3.3 | | |
| Isopoda | | | 0.5 | 5.3 | | | | | | | | |
| Insecta | 88.2 | 80.0 | 25.2 | 44.7 | | | 1.9 | 10.3 | | | 99.5 | 100.0 |
| Insecta (larvae) | | | 1.4 | 13.2 | | | | | | | | |
| Pisces | | | 0.2 | 2.6 | | | | | 0.1 | 3.3 | | |
| Varia | | | | | | | | | 6.0 | 56.7 | 0.5 | 18.2 |
| TROPH ± SE | | 3.19 ± 0.39 | | 3.08 ± 0.31 | | | | 3.00 ± 0.00 | | 3.09 ± 0.24 | | 3.20 ± 0.40 |
| TROPH ± SE [28] | | | | | | | | | | | | 3.20 ± 0.36 |
| TROPH [4] | | | | | | | | | | | | 3.30 |
| *Boops boops* Individuals/Juveniles | 2 | 2 | 30 | 30 | 30 (1) | 30 | 30 (4) | 29 | 30 (1) | 30 | 30 | 30 |
| Mean TL (mm)/mean wet weight (g) | | 78.50/3.5 | | 67.00/1.8 | | 87.81/6.2 | | 103.07/9.8 | | 100.89/9.4 | | 94.02/6.7 |
| Algae | | | + | + | | | | | | + | | |
| Gastropoda | | | | | | | | | | | 0.12 | 3.3 |
| Bivalvia | | | | | | | | | | | 0.12 | 3.3 |
| Polychaeta | 25.0 | 50.0 | 0.9 | 33.3 | 0.3 | 6.9 | | | 0.1 | 10.3 | 0.12 | 3.3 |
| Crustacea (larvae) | 50.0 | 100.0 | 84.6 | 96.7 | 29.4 | 82.8 | 4.1 | 30.8 | 3.9 | 69.0 | 3.25 | 46.7 |
| Copepoda | | | 12.6 | 56.7 | 45.9 | 44.8 | 79.4 | 92.3 | 90.3 | 96.6 | 40.4 | 90.0 |
| Ostracoda | | | 0.3 | 10.0 | | | | | 0.1 | 10.3 | 0.35 | 10.0 |
| Decapoda | 25.0 | 50.0 | | | | | | | <0.1 | 3.4 | 0.12 | 3.3 |
| Tanaidacea | | | | | | | | | 0.1 | 6.9 | 0.12 | 3.3 |
| Isopoda | | | 0.3 | 6.7 | | | 0.3 | 3.8 | | | | |
| Amphipoda | | | 0.1 | 6.7 | | | | | 0.1 | 3.4 | 1.28 | 26.7 |
| Pycnogonida | | | | | | | | | <0.1 | 3.4 | | |
| Insecta | | | | | | | 0.6 | 3.8 | | | 1.28 | 16.7 |
| Insecta (larvae) | | | | | | | 0.3 | 3.8 | <0.1 | 3.4 | | |
| Pisces | | | 0.3 | 13.3 | 16.4 | 44.8 | 0.3 | 3.8 | | | 1.97 | 33.3 |
| Varia | | | 0.9 | 16.7 | 8.1 | 24.1 | 15.0 | 42.3 | 5.3 | 48.3 | 50.9 | 80.0 |

| Taxonomic Prey Categories | May '06 | | July '06 | | Sept '06 | | Nov '06 | | Feb '07 | | Apr '07 | |
|---|---|---|---|---|---|---|---|---|---|---|---|---|
| | %N | %F | %N | %F | %N | %F | %N | %F | %N | %F | %N | %F |
| TROPH ± SE | 3.17 ± 0.34 | | 3.09 ± 0.27 | | 3.35 ± 0.36 | | 3.09 ± 0.12 | | 3.05 ± 0.08 | | 3.26 ± 0.35 | |
| TROPH ± SE [28] | | | | | | | | | | | 2.80 ± 0.00 | |
| TROPH [4] | | | | | | | | | | | 3.12 | |
| *Diplodus annularis* Individuals/Juveniles | | | 9 (1) | 9 | 6 | 6 | | | 30 | 30 | 1 | 1 |
| Mean TL (mm)/mean wet weight (g) | | | | 53.63/4.3 | | 61.98/3.5 | | | | 62.80/3.2 | | 63.91/4.0 |
| Bivalvia | | | | | | | | | 0.3 | 10.0 | | |
| Polychaeta | | | | | 1.5 | 16.7 | | | 0.2 | 3.3 | | |
| Crustacea (larvae) | | | | | | | | | | | 3.2 | 100.0 |
| Copepoda | | | 98.3 | 100.0 | 32.4 | 66.7 | | | 6.9 | 40.0 | 35.5 | 100.0 |
| Ostracoda | | | | | 5.9 | 16.7 | | | 0.5 | 10.0 | | |
| Mysidacea | | | | | | | | | 0.2 | 3.3 | | |
| Cumacea | | | 0.1 | 12.5 | | | | | | | | |
| Tanaidacea | | | 0.4 | 50.0 | 42.6 | 66.7 | | | 16.4 | 80.0 | 29.0 | 100.0 |
| Isopoda | | | 0.1 | 12.5 | 2.9 | 33.3 | | | 0.2 | 3.3 | | |
| Amphipoda | | | 1.1 | 75.0 | 14.7 | 50.0 | | | 75.3 | 96.7 | 32.3 | 100.0 |
| Varia | | | | | | | | | 0.2 | 3.3 | | |
| TROPH ± SE | | | 3.00 ± 0.00 | | 3.22 ± 0.44 | | | | 3.26 ± 0.50 | | 3.19 ± 0.41 | |
| TROPH ± SE [28] | | | | | | | | | | | 3.60 ± 0.00 | |
| TROPH [4] | | | | | | | | | | | 3.19 | |
| *Pagellus acarne* Individuals/Juveniles | 30 | 30 | 31 (1) | 31 | 22 (5) | 22 | | | 14 | 14 | 30 | 30 |
| Mean TL (mm)/mean wet weight (g) | | 60.07/1.8 | | 65.01/3.3 | | 68.89/3.8 | | | | 86.18/7.3 | | 61.77/4.4 |
| Gastropoda | 0.8 | 26.7 | | | | | | | | | 1.9 | 37.9 |
| Polychaeta | | | | | | | | | 0.3 | 7.1 | | |
| Crustacea (larvae) | 5.7 | 23.3 | <0.1 | 3.3 | | | | | 1.4 | 21.4 | 0.2 | 6.9 |
| Copepoda | 85.2 | 100.0 | 99.5 | 100.0 | 32.7 | 70.6 | | | 17.5 | 28.6 | 78.2 | 100.0 |
| Ostracoda | 0.3 | 10.0 | 0.3 | 33.3 | | | | | 9.0 | 71.4 | 3.0 | 34.5 |
| Mysidacea | 0.1 | 3.3 | | | | | | | | | | |
| Tanaidacea | 3.3 | 56.7 | 0.1 | 20.0 | | | | | 16.6 | 100.0 | 4.5 | 51.7 |
| Isopoda | 0.5 | 20.0 | <0.1 | 3.3 | | | | | 0.3 | 7.1 | | |
| Amphipoda | 3.1 | 46.7 | <0.1 | 6.7 | | | | | 48.5 | 100.0 | 6.7 | 41.4 |
| Insecta (larvae) | 1.0 | 3.3 | | | 67.3 | 82.4 | | | 6.5 | 21.4 | 0.1 | 3.4 |

| Taxonomic Prey Categories | May '06 | | July '06 | | Sept '06 | | Nov '06 | | Feb '07 | | Apr '07 | |
|---|---|---|---|---|---|---|---|---|---|---|---|---|
| | %N | %F | %N | %F | %N | %F | %N | %F | %N | %F | %N | %F |
| Pisces | | | <0.1 | 3.3 | | | | | | | | |
| Varia | | | | | | | | | | | 5.5 | 24.1 |
| TROPH ± SE | 3.09 ± 0.16 | | 3.00 ± 0.00 | | 3.14 ± 0.32 | | | | 3.22 ± 0.45 | | 3.04 ± 0.17 | |
| TROPH ± SE [28] | | | | | | | | | | | 3.80 ± 0.00 | |
| TROPH [4] | | | | | | | | | | | 3.61 | |
| *Spicara smaris* Individuals/Juveniles | 30 | 30 | 31 | 31 | 30 | 30 | 30 (1) | 21 | 30 | 19 | 29 (1) | 26 |
| Mean TL (mm)/mean wet weight (g) | | 64.74/2.2 | | 44.58/1.7 | | 77.98/4.7 | | 86.75/7.5 | | 88.74/5.2 | | 82.58/4.4 |
| Gastropoda | 1.3 | 13.3 | | | | | | | | | 0.1 | 3.6 |
| Polychaeta | | | | | | | 0.1 | 3.4 | | | 0.1 | 3.6 |
| Crustacea (larvae) | 7.5 | 53.3 | 0.1 | 6.5 | 0.6 | 40.0 | 0.7 | 17.2 | <0.1 | 3.3 | 0.1 | 3.6 |
| Copepoda | 57.7 | 96.7 | 99.5 | 100.0 | 99.1 | 100.0 | 97.6 | 96.6 | 99.4 | 100.0 | 50.5 | 71.4 |
| Ostracoda | | | 0.4 | 16.1 | | | | | | | | |
| Decapoda | | | < 0.1 | 3.2 | | | | | | | | |
| Tanaidacea | | | | | | | | | | | 0.1 | 3.6 |
| Isopoda | 0.5 | 10.0 | | | | | 0.1 | 3.4 | | | | |
| Amphipoda | 0.2 | 3.3 | | | | | 0.1 | 3.4 | | | | |
| Insecta (larvae) | 32.8 | 76.7 | | | | | 0.1 | 3.4 | 0.4 | 23.3 | | |
| Pisces | | | | | 0.3 | 16.7 | | | | | 0.1 | 3.6 |
| Varia | | | | | | | 1.2 | 24.1 | 0.2 | 26.7 | 49.1 | 89.3 |
| TROPH ± SE | 3.12 ± 0.26 | | 3.00 ± 0.00 | | 3.00 ± 0.00 | | 3.00 ± 0.00 | | 3.00 ± 0.00 | | 3.19 ± 0.29 | |
| TROPH ± SE [28] | | | | | | | | | | | 3.00 ± 0.00 | |
| TROPH [4] | | | | | | | | | | | 3.20 | |
| *Spicara maena* Individuals/Juveniles | | | | | | | 30 | 29 | 30 (4) | 30 | 1 | 1 |
| Mean TL (mm)/mean wet weight (g) | | | | | | | | 93.66/8.4 | | 93.00/8.0 | | 82.47/5.0 |
| Gastropoda | | | | | | | | | <0.1 | 7.7 | | |
| Polychaeta | | | | | | | 0.4 | 6.7 | | | | |
| Crustacea (larvae) | | | | | | | 0.8 | 13.3 | 0.3 | 38.5 | | |
| Copepoda | | | | | | | 93.7 | 100.0 | 99.4 | 100.0 | 62.5 | 100.0 |
| Ostracoda | | | | | | | | | | | 12.5 | 100.0 |
| Tanaidacea | | | | | | | | | <0.1 | 3.8 | 12.5 | 100.0 |
| Isopoda | | | | | | | 1.2 | 3.3 | | | | |
| Amphipoda | | | | | | | 0.2 | 3.3 | <0.1 | 7.7 | 12.5 | 100.0 |

| Taxonomic Prey Categories | May '06 | | July '06 | | Sept '06 | | Nov '06 | | Feb '07 | | Apr '07 | |
|---|---|---|---|---|---|---|---|---|---|---|---|---|
| | %N | %F | %N | %F | %N | %F | %N | %F | %N | %F | %N | %F |
| Insecta | | | | | | | | | <0.1 | 3.8 | | |
| Insecta (larvae) | | | | | | | | | 0.1 | 3.8 | | |
| Varia | | | | | | | 3.8 | 20.0 | 0.1 | 7.7 | | |
| TROPH ± SE | | | | | | | 3.00 ± 0.00 | | 3.00 ± 0.00 | | 3.13 ± 0.31 | |
| TROPH ± SE [28] | | | | | | | | | | | 4.20 ± 0.60 | |
| TROPH [4] | | | | | | | | | | | 3.25 | |
| Gobius niger Individuals | 30 | 11–25 | 30 | 8–24 | 30 | 5–26 | 30 | 7–26 | 27 | 8–25 | 30 | 4–19 |
| Mean TL (mm)/mean wet weight (g) | | 63.36/2.3 | | 64.51/2.6 | | 66.83/4.0 | | 62.94/1.9 | | 61.12/2.4 | | 76.82/4.0 |
| Algae | | | + | + | | | | | | | | |
| Sipuncula | 0.6 | 3.4 | | | | | | | | | | |
| Polyplacophora | | | | | | | 1.4 | 6.9 | | | | |
| Gastropoda | 3.5 | 20.7 | | | 2.8 | 13.3 | 0.7 | 3.4 | 0.5 | 3.7 | 1.0 | 10.7 |
| Bivalvia | 3.5 | 20.7 | 4.0 | 23.3 | 2.8 | 16.7 | 3.4 | 13.8 | | | 3.8 | 28.6 |
| Polychaeta | 2.9 | 17.2 | 14.9 | - | 9.5 | 46.7 | 7.5 | 31.0 | 5.2 | 37.0 | 1.4 | 14.3 |
| Crustacea (larvae) | | | | | 0.6 | 3.3 | | | | | | |
| Copepoda | 2.3 | 6.9 | 9.2 | | | | 1.4 | 6.9 | 4.6 | 25.9 | 1.0 | 10.7 |
| Ostracoda | 0.6 | 3.4 | | | | | | | | | | |
| Decapoda | 0.6 | - | 1.7 | 10.0 | 2.2 | 13.3 | 1.4 | 6.9 | 1.0 | 7.4 | 1.0 | 10.7 |
| Mysidacea | | | | | | | | | 0.5 | 3.7 | | |
| Cumacea | | | 0.6 | 3.3 | | | 1.4 | 6.9 | | | 0.7 | 7.1 |
| Tanaidacea | | | 4.6 | 26.7 | 0.6 | 3.3 | 4.1 | 10.3 | 2.6 | 14.8 | 2.4 | 14.3 |
| Isopoda | 1.2 | 6.9 | 2.3 | 13.3 | 1.1 | 6.7 | 1.4 | 6.9 | 1.5 | 11.1 | | |
| Amphipoda | 0.6 | 3.4 | 9.2 | 33.3 | 1.1 | 6.7 | 9.6 | 37.9 | 25.8 | 70.4 | 4.8 | 25.0 |
| Insecta (larvae) | 78.5 | 58.6 | 35.6 | 43.3 | 78.8 | 83.3 | 63.7 | 72.4 | 55.2 | 88.9 | 74.7 | 82.1 |
| Pisces | 0.6 | 3.4 | | | 0.6 | 3.3 | 3.4 | 13.8 | 0.5 | 3.7 | 1.0 | 10.7 |
| Varia | 5.2 | 13.8 | 17.8 | 30.0 | | | 0.7 | 3.4 | 2.6 | 14.8 | 8.0 | 25.0 |
| TROPH ± SE | 3.26 ± 0.42 | | 3.24 ± 0.40 | | 3.22 ± 0.40 | | 3.25 ± 0.43 | | 3.21 ± 0.43 | | 3.25 ± 0.41 | |
| TROPH ± SE [28] | | | | | | | | | | | 3.30 ± 0.20 | |
| TROPH [4] | | | | | | | | | | | 3.46 | |
| *Mullus barbatus* Individuals/Juveniles | 30 | 29 | 30 | 30 | 30 | 22 | 30 (1) | 30 | 30 (1) | 30 | | |
| Mean TL (mm)/mean wet weight (g) | | 93.83/8.0 | | 72.02/2.3 | | 98.59/7.6 | | 70.62/3.0 | | 72.8/4.7 | | |

| Taxonomic Prey Categories | May '06 | | July '06 | | Sept '06 | | Nov '06 | | Feb '07 | | Apr '07 | |
|---|---|---|---|---|---|---|---|---|---|---|---|---|
| | %N | %F | %N | %F | %N | %F | %N | %F | %N | %F | %N | %F |
| Mollusca | | | 0.1 | 6.7 | | | | | | | | |
| Gastropoda | 0.8 | 3.3 | | | | | 5.7 | 13.8 | 0.7 | 6.9 | | |
| Bivalvia | 17.6 | 63.3 | 0.1 | 10.0 | 12.2 | 10.0 | 3.1 | 17.2 | 5.3 | 37.9 | | |
| Polychaeta | 4.2 | 16.7 | 1.1 | 56.7 | 22.2 | 60.0 | 4.1 | 44.8 | 2.6 | 20.7 | | |
| Copepoda | 5.9 | 10.0 | 91.2 | 80.0 | 24.9 | 30.0 | 41.5 | 72.4 | 59.3 | 86.2 | | |
| Ostracoda | | | 0.1 | 6.7 | 0.5 | 3.3 | | | | | | |
| Decapoda | 3.4 | 13.3 | 0.2 | 16.7 | 4.2 | 13.3 | 3.1 | 20.7 | | | | |
| Mysidacea | 7.6 | 30.0 | 0.1 | 10.0 | | | 0.3 | 3.4 | 0.6 | 13.8 | | |
| Cumacea | 6.7 | 20.0 | 0.1 | 10.0 | 4.8 | 23.3 | 8.5 | 51.7 | 7.7 | 69.0 | | |
| Tanaidacea | 5.0 | 20.0 | 0.3 | 30.0 | 2.6 | 16.7 | 1.3 | 13.8 | 1.1 | 24.1 | | |
| Isopoda | 5.9 | 23.3 | 1.8 | 63.3 | 6.3 | 23.3 | 1.8 | 20.7 | 1.0 | 20.7 | | |
| Amphipoda | 39.5 | 63.3 | 4.8 | 90.0 | 12.2 | 33.3 | 21.1 | 86.2 | 21.1 | 89.7 | | |
| Pycnogonida | | | < 0.1 | 3.3 | | | | | | | | |
| Insecta (larvae) | 0.8 | 3.3 | | | 0.5 | 3.3 | 0.3 | 3.4 | 0.1 | 3.4 | | |
| Pisces | | | 0.2 | 20.0 | 3.7 | 16.7 | | | | | | |
| Varia | 2.5 | 10.0 | | | 5.8 | 23.3 | 9.3 | 41.4 | 0.4 | 10.3 | | |
| TROPH ± SE | 3.23 ± 0.47 | | 3.03 ± 0.12 | | 3.18 ± 0.34 | | 3.14 ± 0.29 | | 3.08 ± 0.24 | | | |
| TROPH ± SE [28] | | | | | | | | | | | 3.10 ± 0.10 | |
| TROPH [4] | | | | | | | | | | | 3.23 | |
| *Mullus surmuletus* Individuals/Juveniles | | | 30 | 30 | 7 | 7 | | | 4 | 3 | | |
| Mean TL (mm)/mean wet weight (g) | | | | 85.18/3.9 | | 111.42/17.6 | | | | 131.94/34.8 | | |
| Gastropoda | | | 0.2 | 10.0 | | | | | | | | |
| Polychaeta | | | 0.4 | 20.0 | 11.8 | 33.3 | | | | | | |
| Copepoda | | | 88.8 | 46.7 | 17.6 | 33.3 | | | | | | |
| Ostracoda | | | 0.1 | 3.3 | | | | | | | | |
| Decapoda | | | 0.4 | 23.3 | 41.2 | 50.0 | | | 57.1 | 75.0 | | |
| Mysidacea | | | 0.3 | 16.7 | | | | | | | | |
| Cumacea | | | 0.1 | 3.3 | | | | | | | | |
| Tanaidacea | | | 1.9 | 36.7 | | | | | 3.6 | 25.0 | | |
| Isopoda | | | 1.7 | 43.3 | | | | | | | | |
| Amphipoda | | | 4.9 | 80.0 | 23.5 | 50.0 | | | 21.4 | 75.0 | | |
| Insecta (larvae) | | | | | | | | | 17.9 | 25.0 | | |
| Pisces | | | 1.1 | 50.0 | 5.9 | 16.6 | | | | | | |

| Taxonomic Prey Categories | May '06 | | July '06 | | Sept '06 | | Nov '06 | | Feb '07 | | Apr '07 | |
|---|---|---|---|---|---|---|---|---|---|---|---|---|
| | %N | %F | %N | %F | %N | %F | %N | %F | %N | %F | %N | %F |
| TROPH ± SE | | | 3.03 ± 0.11 | | 3.37 ± 0.48 | | | | 3.46 ± 0.55 | | | |
| TROPH ± SE [28] | | | | | | | | | | | 3.50 ± 0.30 | |
| TROPH [4] | | | | | | | | | | | 3.34 | |
| *Parablennius tentacularis* Individuals | 30 | 17–30 | 30 | 10–30 | 6 | 4–6 | 24 | 22–24 | 6 | 5–6 | 30 | 19–30 |
| Mean TL (mm)/mean wet weight (g) | | 72.93/3.4 | | 78.07/4.9 | | 68.05/3.3 | | 59.97/2.2 | | 68.35/3.0 | | 72.74/3.3 |
| Algae | + | + | + | + | + | + | + | + | + | + | + | + |
| Sipuncula | 2.2 | 3.4 | | | | | | | | | | |
| Gastropoda | 2.2 | 3.4 | 9.7 | 19.0 | 50.0 | 20.0 | | | | | 0.6 | 3.7 |
| Bivalvia | 4.4 | 6.9 | | | | | | | | | 1.8 | 3.7 |
| Polychaeta | 4.4 | 13.8 | 19.4 | 28.6 | | | 1.3 | 9.1 | 1.4 | 16.7 | | |
| Copepoda | | | | | | | 1.3 | 9.1 | | | | |
| Ostracoda | | | 3.2 | 4.8 | | | 0.7 | 4.5 | 1.4 | 16.7 | | |
| Decapoda | | | | | | | | | | | 1.8 | 11.1 |
| Tanaidacea | 6.7 | 10.3 | 9.7 | 14.3 | | | 6.0 | 22.7 | 6.8 | 33.3 | 8.6 | 37.0 |
| Isopoda | 24.4 | 37.9 | 35.5 | 38.1 | 50.0 | 20.0 | 2.7 | 13.6 | | | 1.2 | 7.4 |
| Amphipoda | 35.6 | 27.6 | 9.7 | 14.3 | | | 85.9 | 81.8 | 83.8 | 100.0 | 85.9 | 59.3 |
| Insecta (larvae) | | | 3.2 | 4.8 | | | 0.7 | 4.5 | | | | |
| Varia | 20.0 | 27.6 | 9.7 | 14.3 | | | 1.3 | 9.1 | 6.8 | 16.7 | | |
| TROPH ± SE | 3.34 ± 0.51 | | 3.27 ± 0.48 | | 3.33 ± 0.56 | | 3.29 ± 0.53 | | 3.31 ± 0.52 | | 3.29 ± 0.53 | |
| TROPH ± SE [28] | | | | | | | | | | | 3.30 ± 0.30 | |
| TROPH [4] | | | | | | | | | | | 3.11 | |
| *Serranus hepatus* Individuals/Juveniles | 30 (1) | 30 | 17 (4) | 16 | 30 (2) | 29 | 30 (3) | 28 | 30 | 30 | 30 (5) | 30 |
| Mean TL (mm)/mean wet weight (g) | | 64.42/3.9 | | 69.32/5.4 | | 53.50/2.2 | | 55.57/1.9 | | 51.86/2.0 | | 64.09/3.4 |
| Polychaeta | 1.3 | 3.4 | 2.0 | 7.7 | 5.1 | 10.7 | 0.6 | 3.7 | | | 0.8 | 4.0 |
| Crustacea (larvae) | 1.3 | 3.4 | | | | | | | | | 0.8 | 4.0 |
| Copepoda | 8.8 | 10.3 | 56.0 | 7.7 | 3.4 | 3.6 | 14.6 | 33.3 | 4.3 | 20.0 | 1.5 | 4.0 |
| Ostracoda | | | | | | | 0.6 | 3.7 | | | | |
| Decapoda | 25.0 | 55.2 | 20.0 | 69.2 | 35.6 | 64.3 | 5.1 | 29.6 | 6.4 | 50.0 | 16.9 | 64.0 |
| Mysidacea | 2.5 | 6.9 | | | 15.3 | 28.6 | 0.6 | 3.7 | 1.1 | 10.0 | 0.8 | 4.0 |
| Cumacea | | | | | | | | | 0.4 | 3.3 | 0.8 | 4.0 |
| Tanaidacea | | | | | | | 1.3 | 7.4 | 1.1 | 6.7 | 3.8 | 20.0 |
| Isopoda | 30.0 | 55.2 | | | 13.6 | 25.0 | 2.5 | 11.1 | 2.1 | 13.3 | 1.5 | 8.0 |
| Amphipoda | 17.5 | 27.6 | 14.0 | 23.1 | 23.7 | 39.3 | 71.5 | 81.5 | 82.9 | 100.0 | 58.5 | 52.0 |

| Taxonomic Prey Categories | May '06 | | July '06 | | Sept '06 | | Nov '06 | | Feb '07 | | Apr '07 | |
|---|---|---|---|---|---|---|---|---|---|---|---|---|
| | %N | %F | %N | %F | %N | %F | %N | %F | %N | %F | %N | %F |
| Insecta (larvae) | | | | | | | | | | | 1.5 | 4.0 |
| Varia | 12.5 | 27.6 | 8.0 | 30.8 | 3.4 | 7.1 | 1.3 | 7.4 | | | 13.1 | 20.0 |
| TROPH ± SE | 3.43 ± 0.52 | | 3.22 ± 0.34 | | 3.44 ± 0.54 | | 3.26 ± 0.50 | | 3.31 ± 0.54 | | 3.35 ± 0.54 | |
| TROPH ± SE [28] | | | | | | | | | | | 3.50 ± 0.40 | |
| TROPH [4] | | | | | | | | | | | 3.63 | |
| *Symphodus cinereus* Individuals/Juveniles | | | 21 | 0 | 31 (3) | 0 | 39 (1) | 0 | 11 | 0 | | |
| Mean TL (mm)/mean wet weight (g) | | | | 70.11/5.6 | | 70.90/5.0 | | 78.11/6.3 | | 82.17/7.6 | | |
| Gastropoda | | | | | 0.7 | 3.6 | 1.2 | 2.6 | 0.4 | 9.1 | | |
| Bivalvia | | | | | 2.0 | 3.6 | 0.6 | 5.3 | 2.2 | 27.3 | | |
| Polychaeta | | | 1.8 | 19.0 | 2.7 | 10.7 | 0.2 | 2.6 | | | | |
| Copepoda | | | 41.8 | 52.4 | 1.4 | 7.1 | 0.6 | 2.6 | 0.7 | 9.1 | | |
| Ostracoda | | | 0.4 | 4.8 | 2.0 | 7.1 | | | | | | |
| Decapoda | | | 1.8 | 23.8 | 5.4 | 14.3 | 1.2 | 7.9 | | | | |
| Cumacea | | | | | | | 0.2 | 2.6 | | | | |
| Tanaidacea | | | 16.8 | 76.2 | 29.7 | 60.7 | 17.9 | 71.1 | 10.1 | 81.8 | | |
| Isopoda | | | 3.7 | 28.6 | 7.4 | 35.7 | 1.6 | 21.1 | 0.4 | 9.1 | | |
| Amphipoda | | | 31.9 | 90.5 | 48.0 | 82.1 | 76.5 | 94.7 | 83.8 | 100.0 | | |
| Pisces | | | | | 0.7 | 3.6 | | | | | | |
| Varia | | | 1.8 | 14.3 | | | 0.2 | 2.6 | 2.5 | 36.4 | | |
| TROPH ± SE | | | 3.18 ± 0.41 | | 3.30 ± 0.52 | | 3.30 ± 0.53 | | 3.30 ± 0.53 | | | |
| TROPH ± SE [28] | | | | | | | | | | | 3.50 ± 0.10 | |
| TROPH [4] | | | | | | | | | | | 3.23 | |
| *Pagrus pagrus* Individuals/Juveniles | 2 | 2 | 22 (1) | 22 | 17 (2) | 17 | 4 | 4 | | | 1 | 1 |
| Mean TL (mm)/mean wet weight (g) | | 104.29/3.9 | | 90.27/15.8 | | 107.53/33.6 | | 78.11/16.5 | | | | 142.48/49 |

| Taxonomic Prey Categories | May '06 | | July '06 | | Sept '06 | | Nov '06 | | Feb '07 | | Apr '07 | |
|---|---|---|---|---|---|---|---|---|---|---|---|---|
| | %N | %F | %N | %F | %N | %F | %N | %F | %N | %F | %N | %F |
| Sipuncula | | | 9.4 | 14.3 | 1.5 | 13.3 | | | | | | |
| Polyplacophora | | | | | 1.5 | 13.3 | | | | | | |
| Polychaeta | 9.1 | 50.0 | 6.3 | 9.5 | 2.2 | 20.0 | | | | | | |
| Crustacea larvae | | | | | 67.4 | 6.7 | | | | | | |
| Copepoda | | | | | 7.4 | 6.7 | | | | | | |
| Decapoda | 18.2 | 100.0 | 56.3 | 76.2 | 15.6 | 66.7 | 40.0 | 50.0 | | | 20.0 | 100.0 |
| Mysidacea | | | 3.1 | 4.8 | | | 20.0 | 25.0 | | | | |
| Tanaidacea | | | | | 1.5 | 6.7 | | | | | | |
| Isopoda | | | 12.5 | 14.3 | | | | | | | | |
| Amphipoda | 72.7 | 50.0 | | | | | | | | | | |
| Fish | | | 3.1 | 4.8 | 1.5 | 13.3 | | | | | | |
| Varia | | | 9.4 | 14.3 | 1.5 | 13.3 | 40.0 | 50.0 | | | 80.0 | 100.0 |
| TROPH ± SE | 3.33 ± 0.52 | | 3.46 ± 0.52 | | 3.47 ± 0.49 | | 3.45 ± 0.43 | | | | 3.50 ± 0.40 | |
| TROPH ± SE [28] | | | | | | | | | | | 3.90 ± 0.20 | |
| TROPH [4] | | | | | | | | | | | 3.71 | |

## Appendix B

Averaged percentage numerical abundance (%N) of all the taxa found as prey in the stomachs of fish species examined, also found in the macrobenthic and zooplanktonic samples (MB: macrobenthos; MZ: macrozooplankton), during the six sampling occasions in the study area.

| Taxa | MB | MZ | Dipan | Gobnig | Mulbar | Mulsur | Pagacar | Pagpag | Parten | Serhep | Symcin | Athboy | Bopbop | Spicsmar | Spicmaen |
|---|---|---|---|---|---|---|---|---|---|---|---|---|---|---|---|
| *Phascolosoma* sp. | 0.06 | | | 0.1 | | | | 2.17 | 0.37 | | | | | | |
| Sipuncula | 0.36 | | | 0.1 | | | | 2.17 | 0.37 | | | | | | |
| *Acanthochitona crinita* Pennant, 1777 | 0.09 | | | | | | | 0.15 | | | | | | | |
| *Chiton* (*Rhyssoplax*) *phaseolinus* Monterosato, 1879 | <0.01 | | | 0.23 | | | | 0.15 | | | | | | | |
| Polyplacophora | 0.10 | | | 0.23 | | | | 0.30 | | | | | | | |
| *Bittium* sp. | | | | 0.09 | 0.17 | | | | 0.54 | | | | | | |

| Taxa | MB | MZ | Dipan | Gobnig | Mulbar | Mulsur | Pagacar | Pagpag | Parten | Serhep | Symcin | Athboy | Bopbop | Spicsmar | Spicmaen |
|---|---|---|---|---|---|---|---|---|---|---|---|---|---|---|---|
| Gastropoda (unidentified) | | | | 0.54 | 0.14 | 0.06 | 0.55 | | 1.45 | | 0.55 | | 0.02 | 0.24 | 0.02 |
| *Gibbula* sp. | | | | 0.19 | | | | | | | | | | | |
| *Naticidae* spp. | | | | 0.39 | | | | | | | | | | | |
| *Pyramidellidae* spp. | | | | | 0.05 | | | | | | | | | | |
| *Raphitoma philberti* (Michaud, 1829) | <0.01 | | | | | | | | 8.44 | | | | | | |
| *Retusa umbilicata* (Montagu, 1803) | 1.60 | | | 0.09 | | | | | | | | | | | |
| *Rissoidea* spp. | | | | 0.11 | 1.08 | | | | | | | | | | |
| Gastropoda | 14.89 | 1.27 | | 1.41 | 1.44 | 0.06 | 0.55 | | 10.43 | | 0.55 | | 0.02 | 0.24 | 0.02 |
| *Abra alba* (W. Wood, 1802) | 7.87 | | 0.04 | 2.39 | 7.66 | | | | 0.37 | | | | 0.02 | | |
| Bivalvia (unidentified) | | | 0.04 | | | | | | | | | | | | |
| *Glans trapezia* (Linnaeus, 1767) | 0.05 | | | | | | | | | | 1.19 | | | | |
| *Limaria hians* (Gmelin, 1791) | 0.04 | | | | | | | | 0.37 | | | | | | |
| *Limaria* sp. | | | | 0.09 | | | | | 0.31 | | | | | | |
| *Mimachlamys varia* (Linnaeus, 1758) | 0.08 | | | 0.34 | | | | | | | | | | | |
| *Modiolus barbatus* (Linnaeus, 1758) | 0.17 | | | 0.09 | | | | | | | | | | | |
| Bivalvia | 12.21 | 0.03 | 0.08 | 2.91 | 7.66 | | | | 1.05 | | 1.19 | | 0.02 | | |
| Mollusca (unidentified) | | | | | 0.01 | | | | | | | | | | |
| *Amphinomidae* spp. | | | | | | | | 1.25 | | | | | | | |
| *Capitellidae* spp. | | | | | | | | 0.15 | | | | | | | |
| *Ceratonereis* (*Composetia*) *vittata* Langerhans, 1884 | 0.16 | | | 0.48 | | | | 0.15 | 0.54 | 0.21 | 0.18 | 0.19 | 4.39 | | |
| *Dorvileidae* spp. | | | | 0.10 | 0.38 | | | | | | | | | | |
| *Euclimene* sp. | | | | 1.13 | 0.16 | | | | | | | | | | |
| *Eunicidae* spp. | | | | 1.64 | 3.91 | 0.02 | | | | | | 0.69 | 0.02 | | |
| *Glycera alba* (O.F. Müller, 1776) | 0.22 | | | 0.77 | 0.58 | | | | | | | | | | |
| *Hesionidae* spp. | | | | 0.83 | 0.11 | | 0.06 | 0.15 | | | | | | | 0.13 |
| *Lumbrineridae* spp. | | | | 0.23 | | | | | | 0.28 | | | | | |
| *Nephthydae* spp. | | | | | | | | 1.82 | | | | | | | |
| *Pectinaridae* spp. | | | | 0.48 | 0.55 | | | | | | | | | | |
| Polychaeta (unidentified) | | | | 1.15 | 0.32 | | | | 0.85 | | 0.31 | | | | |
| *Pontogenia chrysocoma* (Baird, 1865) | 0.02 | | | | | 4.03 | | | | | 0.74 | | | 0.03 | |
| *Potamilla* sp. | | | | | | | | | | | 0.39 | | | | |
| *Syllidae* spp. | | | | 0.09 | 0.82 | | | | | | | | | | |
| *Vermiliopsis infundibulum* (Philippi, 1844) | 0.01 | | 0.41 | | | | | | 3.03 | | | | | | |

| Taxa | MB | MZ | Dipan | Gobnig | Mulbar | Mulsur | Pagacar | Pagpag | Parten | Serhep | Symcin | Athboy | Bopbop | Spicsmar | Spicmaen |
|---|---|---|---|---|---|---|---|---|---|---|---|---|---|---|---|
| Polychaeta | 6.72 | 0.43 | 0.41 | 6.90 | 6.83 | 4.05 | 0.06 | 3.52 | 4.42 | 1.62 | 1.18 | 0.21 | 4.39 | 0.03 | 0.13 |
| Crustacea (larvae) | 0.02 | 0.63 | 0.81 | 0.09 | | | 1.45 | 13.48 | | 0.34 | | 13.33 | 29.20 | 1.51 | 0.36 |
| Copepoda (calanoids, cyclopoids, cladocerans) | | 46.87 | 43.25 | 3.09 | 44.55 | 35.49 | 62.61 | 1.48 | 0.22 | 15.14 | 11.10 | 41.75 | 44.76 | 83.96 | 84.83 |
| Ostracoda | 0.10 | 0.04 | 1.59 | 0.10 | 0.12 | 0.04 | 2.51 | | 0.87 | 0.11 | 0.60 | 0.01 | 0.12 | 0.06 | 4.17 |
| *Achaeus gracilis* (Costa, 1839) | 0.01 | | | | 0.01 | 1.96 | | | | | | | | | |
| *Alpheus dentipes* Guérin, 1832 | 0.10 | | | | 0.21 | | | | | 0.21 | | | | | |
| *Athanas nitescens* (Leach, 1813 [in Leach, 18131814]) | 0.18 | | | | 0.21 | 3.94 | | | | 1.21 | | | | | |
| *Clibanarius erythropus* (Latreille, 1818) | 0.03 | | | 0.10 | | | | | | | | | | | |
| Decapoda (unidentified) | | | | 0.10 | 0.11 | | | 4.00 | | 1.67 | | | | 0.01 | 0.01 |
| *Eualus cranchii* (Leach, 1817 [in Leach, 18151875]) | 0.25 | | | | | 21.03 | | | | 1.16 | | | | | |
| *Galathea bolivari* Zariquiey Álvarez, 1950 | 0.03 | | | | 0.01 | | | 1.04 | | 0.56 | | | | | |
| *Hippolyte leptocerus* (Heller, 1863) | 0.27 | | | | 0.11 | 5.88 | | | | 1.83 | | | 4.17 | | |
| *Hippolyte* spp. | | | | | 0.22 | 0.02 | | | | 0.67 | | | | | |
| *Liocarcinus navigator* (Herbst, 1794) | 0.03 | | | 0.21 | | | | 7.64 | | 0.68 | 0.18 | | | | |
| *Paguridae* spp. | | | | | 0.17 | 0.02 | | | | | | | | | |
| *Paguristes syrtensis* de Saint Laurent, 1971 | 0.14 | | | 0.83 | | | | 17.03 | 0.31 | 6.71 | 1.92 | | | | |
| *Palaemon serratus* (Pennant, 1777) | 0.01 | | | | | | | 0.30 | | 2.06 | | | | | |
| *Pillumnus hirtellus* (Linnaeus, 1761) | 0.01 | | | 0.09 | 0.01 | | | | | | | | | | |
| *Processa macrophthalma* Nouvel & Holthuis, 1957 | 0.02 | | | | 0.80 | 0.06 | | | | 1.06 | | | 0.02 | | |
| *Upogebia pusilla* (Petagna, 1792) | <0.01 | | | | 0.33 | | | | | 0.33 | | | | | |
| Decapoda | 1.21 | | | 1.33 | 2.19 | 32.91 | | 30.01 | | 18.15 | 2.1 | | 4.2 | 0.04 | |
| *Anchialina agilis* (Sars G.O., 1877) | 0.16 | | | | 0.04 | | | | | | | | | | |
| *Diamysis* sp. | 1.45 | | | 0.09 | 0.06 | 0.11 | | 4.63 | | 2.95 | | | | | 0.39 |
| *Haplostylus lobatus* (Nouvel, 1951) | 0.12 | 0.04 | | | 1.60 | | 0.02 | | | 0.42 | | | | | |
| Mysida | 1.87 | 0.04 | | 0.09 | 1.70 | 0.11 | 0.02 | 4.63 | | 3.37 | | | | | 0.39 |
| *Cumella (Cumella) limicola* Sars, 1879 | 0.83 | | | | 0.45 | | | | | | | | | | |
| *Iphinoe serrata* Norman, 1867 | 0.70 | 0.02 | | 0.27 | 3.45 | 0.02 | | | | 0.06 | 0.05 | | | | |
| *Iphinoe* sp. | 1.50 | | | 0.17 | 1.66 | | | | | 0.13 | | | | | |
| Cumacea | 3.08 | | 0.02 | 0.44 | 5.56 | 0.02 | | | | 0.19 | 0.05 | | | | |

| Taxa | MB | MZ | Dipan | Gobnig | Mulbar | Mulsur | Pagacar | Pagpag | Parten | Serhep | Symcin | Athboy | Bopbop | Spicsmar | Spicmaen |
|---|---|---|---|---|---|---|---|---|---|---|---|---|---|---|---|
| *Apseudoidea* spp. | 0.02 | | | | 1.24 | 0.26 | | | | | 0.37 | | | | |
| *Leptochelia* sp. | 11.02 | | 22.12 | 2.38 | 0.84 | 1.55 | 4.90 | 0.30 | 6.29 | 1.03 | 18.27 | | 0.03 | 0.01 | 4.17 |
| Tanaidacea | 11.55 | | 22.12 | 2.38 | 2.08 | 1.81 | 4.90 | 0.30 | 6.29 | 1.03 | 18.64 | | 0.03 | 0.01 | 4.17 |
| *Anthuridea* spp. | 0.34 | | | | | | | | | | | | 0.03 | | |
| *Astacilla* sp. | 0.09 | | 0.04 | 0.11 | | | | | 1.82 | 1.46 | 0.34 | 0.05 | | | |
| *Cymodoce truncata* Leach, 1814 | 0.31 | | 0.74 | 0.83 | 1.28 | 0.13 | 0.02 | 2.50 | 16.68 | 6.10 | 1.88 | | | | |
| *Eurydice* spp. | <0.01 | | | | | | 0.06 | | | 0.16 | 0.22 | | 0.08 | 0.02 | 0.39 |
| *Gnathia dentata* (Sars G.O., 1872) | 0.91 | | 0.02 | 0.30 | 1.96 | 0.45 | 0.09 | | 0.37 | | 0.81 | 0.05 | | | 0.08 |
| *Ianiropsis breviremis* (Sars, 1883) | 0.67 | | | | 0.13 | | | | 0.10 | 0.56 | | | | | |
| Isopoda | 2.35 | | 0.8 | 1.24 | 3.37 | 0.58 | 0.17 | 2.50 | 18.97 | 8.28 | 3.25 | 0.10 | 0.11 | 0.10 | 0.39 |
| *Ampelisca* spp. | <0.01 | | | 0.10 | | | | | | | | | | | |
| *Amphipoda* (unidentified) | | | | | 0.05 | | | | | | | | 0.03 | | 0.01 |
| *Amphithoe ramondi* Audouin, 1826 | 0.05 | | | | | | | | | 0.11 | | | | | |
| *Aora spinicornis* Afonso, 1976 | 2.08 | | 6.74 | 3.81 | | 1.19 | 2.44 | | 26.99 | 3.56 | 31.56 | | | | |
| *Aoridae* spp. | | | 8.37 | 0.77 | 0.18 | | 0.86 | | 2.47 | 0.42 | 2.14 | | | | |
| *Apherusa bispinosa* (Bate, 1857) | 0.58 | | | | 0.01 | 1.19 | | | | | 0.14 | | | | |
| *Autonoe spiniventris* Della Valle, 1893 | 0.12 | | | 0.06 | | | | | | | | | | | |
| *Caprella acanthifera discrepans* Mayer, 1890 | 3.26 | | 0.30 | 1.88 | 0.67 | 0.06 | | | 4.46 | 5.62 | 4.37 | | | | |
| *Caprella acanthifera acanthifera* Leach, 1814 | 6.86 | | 3.36 | 0.35 | 0.06 | | 0.67 | 5.45 | 7.95 | 4.24 | 0.64 | | 0.05 | | 4.17 |
| *Caprella rapax* Mayer, 1890 | | | 0.98 | | 0.43 | 0.13 | 0.26 | 7.27 | 1.45 | 7.08 | 0.43 | | | | |
| *Caprellidae* spp. | | | 0.06 | 0.09 | 0.15 | | 0.30 | | 0.60 | 0.99 | 0.24 | | | | |
| *Dexamine spinosa* (Montagu, 1813) | 0.27 | | | | 0.01 | | | | | | | | | | |
| *Ericthonius punctatus* (Bate, 1857) | 0.72 | | 0.08 | | | | | | | | | | | | |
| *Ericthonius* sp. | | | 3.91 | 0.29 | 0.05 | | 4.50 | | 2.13 | 1.37 | 2.94 | | | | |
| *Guernea* (*Guernea*) *coalita* (Norman. 1868) | 0.07 | | | | 0.03 | | | | | | | | | | |
| *Leptocheirus bispinosus* Norman, 1908 | 1.99 | | | 0.06 | 0.81 | 0.02 | | | 0.10 | 0.11 | 1.37 | | | | |
| *Leptocheirus pectinatus* (Norman, 1869) | 1.06 | | | | 0.22 | | | | | | | | | | |
| *Leptocheirus* sp. | | | | | 1.63 | 0.02 | | | | | 1.46 | | | | |
| *Leucothoe spinicarpa* (Abildgaard, 1789) | 0.39 | | 0.04 | | | | | | 0.74 | 0.13 | | | | | |
| *Lysianassa caesarea* Ruffo, 1987 | 0.35 | | | | | 0.04 | | | | 0.56 | | | | | |
| *Lysianassa pilicornis* (Heller, 1866) | 0.60 | | | | | 3.30 | | | | 1.04 | | | | | |

| Taxa | MB | MZ | Dipan | Gobnig | Mulbar | Mulsur | Pagacar | Pagpag | Parten | Serhep | Symcin | Athboy | Bopbop | Spicsmar | Spicmaen |
|---|---|---|---|---|---|---|---|---|---|---|---|---|---|---|---|
| *Lysianassa* spp. | | | | | 0.68 | | | | 0.37 | | | | | | |
| *Lysianassidae* spp. | | | | 0.09 | 0.26 | 0.17 | | | | | 0.17 | | | | |
| *Lysianassina longicornis* (Lucas, 1846) | 0.06 | | | | 0.11 | | | | | | | | | | |
| *Megaluropus massiliensis* Ledoyer, 1976 | 0.69 | | | | 0.19 | | | | | | | | | | |
| *Microdeutopus anomalus* (Rathke, 1843) | 1.95 | | | | 0.36 | 0.09 | | | | | 0.71 | | | | |
| *Microdeutopus bifidus* Myers, 1977 | 3.51 | | 1.18 | | 1.85 | 0.06 | | | 0.85 | 0.95 | 1.86 | | | | |
| *Microdeutopus* sp. | | | | 0.21 | 1.24 | 0.04 | | | 0.11 | 0.49 | 1.10 | | | | |
| *Microdeutopus stationis* Della Valle, 1893 | 5.35 | 4.36 | 0.12 | 4.45 | 7.96 | 1.19 | 1.82 | | 0.45 | 3.12 | 6.93 | | 0.04 | 0.05 | 0.07 |
| *Microdeutopus versiculatus* (Bate, 1856) | 3.00 | | 0.53 | 0.42 | 2.38 | 2.09 | 0.09 | | 0.34 | 0.39 | 3.55 | | 0.04 | | |
| *Oedicoretidae* spp. | | | | 0.06 | 0.06 | | | | | | | | | | |
| *Orchomene grimaldii* Chevreux, 1890 | 0.13 | | | | 0.21 | | | | | | | | | | |
| *Orchomene* sp. | | | | | 0.06 | | | | | | | | | | |
| *Perioculodes longimanus longimanus* (Bate & Westwood, 1868) | 0.79 | | | 0.12 | 3.30 | 0.23 | 0.06 | | | | | | | 0.02 | |
| *Phtisica marina* Slabber, 1769 | 1.90 | | 0.94 | 0.12 | 0.30 | | 1.29 | | 1.02 | 12.09 | 0.60 | | | 0.07 | |
| *Pseudolirius kroyeri* (Haller, 1897) | 0.82 | | | | | | | | | 0.51 | | | | | |
| *Quadrimaera inaequipes* (A. Costa, 1851) | 2.67 | | | | | 0.02 | | | 0.10 | 0.85 | 0.52 | | | | |
| Amphipoda | 44.07 | 0.07 | 30.85 | 8.55 | 19.75 | 16.61 | 11.66 | 14.54 | 50.13 | 44.34 | 60.02 | | 0.25 | 0.05 | 4.25 |
| Pycnogonida | <0.01 | | | | 0.01 | | | | | | | | 0.01 | | |
| Larvae insects | | | | 64.41 | 0.35 | 5.95 | 14.98 | | 0.65 | 0.26 | | 0.28 | 0.06 | 5.55 | 0.02 |
| Insects | | | | | | | | | | | | 42.96 | 0.31 | | 0.01 |
| Pisces | 0.07 | 0.08 | | 1.02 | 0.78 | 2.34 | <0.01 | 0.92 | | 6.37 | 0.17 | 0.06 | 3.16 | 0.06 | |
| Varia | | 0.09 | 0.04 | 5.71 | 3.61 | | 1.10 | 26.17 | 6.30 | 0.82 | 1.14 | 1.31 | 13.37 | 8.41 | 1.26 |
| **Total % percentage** | **98.60** | **49.51** | **100** | **100** | **100** | **100** | **100** | **100** | **100** | **100** | **100** | **100** | **100** | **100** | **100** |
| **(number of taxa)** | **(71)** | **(9)** | **(26)** | **(56)** | **(69)** | **(36)** | **(23)** | **(23)** | **(36)** | **(51)** | **(35)** | **(11)** | **(23)** | **(13)** | **(14)** |

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
