# Peer review of "Trophic Diversity of a Fish Community Associated with a Caulerpa prolifera (Forsskål) Meadow in a Shallow Semi-Enclosed Embayment"

_jmse, doi:10.3390/jmse9020165_

Round 1

Reviewer 1 Report

Introduction

L60-61 is there some references about Caulerpa prolifera impact on biota other than fish. If so, please add some references, information. What is that impact on invertebrates, for example.

Methods

There is a lot of samples, fish and food. It is very good base to do well examinations.

Results

I think the capacity of text is too high. Many of data are shown in figures and tables. Please leave in text only much important data. I think that at least a half of results text can be deleted.

Discussion and conclusion

No comments. It is very well written.

Author Response

Response to Reviewer 1

Introduction

L60-61 is there some references about Caulerpa prolifera impact on biota other than fish. If so, please add some references, information. What is that impact on invertebrates, for example.

Relevant text was added in order to clarify this issue in the last paragraph of the Introduction section.

Methods

There is a lot of samples, fish and food. It is very good base to do well examinations.

Results

I think the capacity of text is too high. Many of data are shown in figures and tables. Please leave in text only much important data. I think that at least a half of results text can be deleted.

Results section was shortened. Only much important data were left in the text of Results section.

Discussion and conclusion

No comments. It is very well written.

Reviewer 2 Report

An interesting paper and one that could provide valuable information on fish diet in this location. Generally the paper was well written, however, there are areas where more information could be given and the presented information clarified. Below are my comments for each section of the paper:

Abstract: 

In the first sentence does your date range refer to spring 2006 through summer 2007? It is unclear. What is the season in question (you say that the study is looking on a seasonal basis)? The second sentence is really awkward and needs to be re-written. The abstract lacks any statistical results. Please include sample sizes, stats, etc. 

Introduction: 

Line 41. Omit Beck et al., 2001

Lines 45-48. Awkward, rewrite

Line 50. Delete "and references therein"

Lines 51, 60, and 63. Grammatical errors

Line 68. You state that an objective is to investigate differences in diet in relation to body size and seasonal changes, but I don't catch where there is seasonal comparison in the study? If there is, it is not well communicated. If you are just looking at diet in relation to one season, or spring and summer, then state that. 

The intro was a good start but could use a lot more information on the local environment and why there is a need to study that particular location and fish community. I don't feel like you answered the 'so what' and 'who cares?' questions. 

Materials and Methods:

Line 80. Put a space before C (for Celcius)

Lines78-81. The only seasonal difference is temperature? If so, I wouldn't include salinity and substrate in the description of seasonal differences. 

Lines 82-82. Define what Ecological Quality Status means exactly. 

Lines 87-88. samples were collected on six occasions-- what were those? days? hauls? total hauls? 

Lines 90-93. Confusing. Re-write. 

Lines 97-98. Fish were weighed after their digestive tracks had been removed? If so, there is a serious flaw in the sampling design. If not, then re-write the methods to reflect the correct order of procedures. 

Lines 98-99. Delete "i.e. the fishes that had not yet reached maturity"-- it's redundant. 

Lines 107-116. Are these data not presented in this paper? Add a column to Table 1 with relative availability of each prey in the environment. 

Lines 145- 146. You state that some species were excluded from analysis because they fed on "fragments of C. proliferation and fishes". Why were these species excluded? I missed the reason for this in the methods or the discussion.

Lines 122-133. Percent occurrence of prey in stomachs determines trophic level? Can you offer further explanation of this? 

Figure 1:

Include a North arrow

What does the asterisks represent? Sampling site? It should be stated in the figure caption. Depth contours should also be stated in the figure caption.

Table 1:

Caption: are these data a summary from all hauls/ sampling events? Include sample sizes of hauls/ capture events. 

How is the table sorted? There doesn't appear to be a logical order. Sort it alphabetically by family or number of individuals high to low.

Is the name of the individual that first described the species necessary to include? If so, be sure to put all in parentheses. 

Why is there a size range for some fish and not for others? Are some averages of all individuals captured or subsamples? 

Results:

Line 154. Delete "Overall"

Lines 172- 181. Run-on sentence, please break up!

Line 227. Insert 'and' between Caprella spp. and A. spinicornis

Line 236. Please give some numbers, relative percentages, something, so the reader does not have to constantly go back to Appendix A and B. 

Lines 237-238. Again, include numbers. And should there be a reference to Appendix A?

Line 241. You assume accidental prey, but there is no discussion on why that should be assumed in the discussion or a citation given in the results to back that up. 

Lines 239- 254. A mix of results and discussion. Separate appropriately. 

Line 250. Missing a parentheses.

Discussion: 

Lines 268 and 269. Commas missing

Line 282. Delete "and references therein"

Lines 286-288. Awkward, rewrite.

Lines 274-291. Smaller sized fish feeding on smaller prey? Unclear

Line 313. What do you mean by "optimum feeding conditions"? Please describe. 

In general, rewrite the discussion so there is less repetition of species by species results and more interpretation of those results-- what does it actually mean? eg lines 322-335 was good discussion material. 

I would have liked to see a lot more discussion on the application of this knowledge. Which of these fish species are commercially or recreationally fished? Which are important prey species to predators in the area? Or provide key food for commercial fish species? What are the current threats to this system and how will this information help mitigate those threats? How unique is this habitat? Can these fish species be used as a proxy for other areas? Were there any surprises from this study? 

Author Response

Response to Reviewer 2

An interesting paper and one that could provide valuable information on fish diet in this location. Generally the paper was well written, however, there are areas where more information could be given and the presented information clarified. Below are my comments for each section of the paper:

Abstract:

In the first sentence does your date range refer to spring 2006 through summer 2007? It is unclear. What is the season in question (you say that the study is looking on a seasonal basis)?

Fish samples were collected on six occasions: May 2006, July 2006, September 2006, November 2006, February 2007 and April 2007. Stomach contents of the fishes were investigated on a bimonthly basis throughout a year (spring 2006-2007). The first sentence was corrected according to the reviewer’s comment.

The second sentence is really awkward and needs to be re-written.

The reviewer is right about the second sentence as it is more or less a conclusion of a previous study in the area and further verifies the conclusions of this study. Therefore this sentence was re-written.

The abstract lacks any statistical results. Please include sample sizes, stats, etc.

Text was added according to the reviewer’s comment. “Overall 1,642 fish individuals, belonging to 17 species, were examined. In total, 45,674 prey individuals were identified to 110 prey taxa, most of which Malacostraca including their larvae and Copepoda (41,175 individuals identified to 71 taxa).”

Introduction:

Line 41. Omit Beck et al., 2001

Beck et al., 2001 was omitted.

Lines 45-48. Awkward, rewrite

The reviewer is right. What we wanted to say by this sentence is that despite the value of feeding ecology and the progress achieved for studying this field through the development of molecular techniques and modelling tools, the traditional approach, i.e. taxonomic identification of prey taxa, is still scanty. However, it comprises basic knowledge and therefore it is essential in order to have a comprehensive view leading to holistic management approaches and conservation strategies.

Text was re-written according to the reviewer’s comment.

Line 50. Delete "and references therein"

Text was deleted according to the reviewer’s comment.

Lines 51, 60, and 63. Grammatical errors

Text was corrected.

Line 68. You state that an objective is to investigate differences in diet in relation to body size and seasonal changes, but I don't catch where there is seasonal comparison in the study? If there is, it is not well communicated. If you are just looking at diet in relation to one season, or spring and summer, then state that.

Comparisons are made, in terms of numerical abundance and frequency of occurrence, between different sampling occasions for the diet of each fish species and shown in Appendix A. Text was corrected according to the reviewer’s comment.

The intro was a good start but could use a lot more information on the local environment and why there is a need to study that particular location and fish community. I don't feel like you answered the 'so what' and 'who cares?' questions.

We agree with the reviewer’s comment and text was corrected and added in the final paragraph of the Introduction section.

Materials and Methods:

Line 80. Put a space before C (for Celcius)

Space was added.

Lines78-81. The only seasonal difference is temperature? If so, I wouldn't include salinity and substrate in the description of seasonal differences.

Salinity and substrate are omitted from seasonal differences.

Lines 82-82. Define what Ecological Quality Status means exactly

Text was modified in order to clarify this issue.

Lines 87-88. samples were collected on six occasions-- what were those? days? hauls? total hauls?

Fish samples were collected on six (days) occasions (one haul sweeping an area of approximately 0.006 km2 on each occasion): May 2006, July 2006, September 2006, November 2006, February 2007 and April 2007 at site A of the study area. Text was corrected in order to clarify this issue.

Lines 90-93. Confusing. Re-write.

Text was re-written in order to clarify this issue.

Lines 97-98. Fish were weighed after their digestive tracks had been removed? If so, there is a serious flaw in the sampling design. If not, then re-write the methods to reflect the correct order of procedures.

The order of the procedure is correct. Fish were weighed before removing their digestive tracks and digestive tracks were also weighed. “After sampling, all individuals were measured to the nearest mm (total length, TL) and weighed to the nearest 0.01g in the laboratory….The stomach and intestine…were then dissected and wet-weighed.”

Lines 98-99. Delete "i.e. the fishes that had not yet reached maturity"-- it's redundant.

Text was deleted according to the reviewer’s comment.

Lines 107-116. Are these data not presented in this paper? Add a column to Table 1 with relative availability of each prey in the environment.

These meta-data are presented in Fig. 3 (major taxonomic groups) and in Appendix B (taxa) of this paper.

Lines 145- 146. You state that some species were excluded from analysis because they fed on "fragments of C. proliferation and fishes". Why were these species excluded? I missed the reason for this in the methods or the discussion.

Siganus luridus was excluded as fragments of C. prolifera cannot be expressed by relative abundance and frerquency of occurrence but only as presence-absence. Shyraena sphyraena was also excluded as all individuals were only from one sampling occasion feeding almost exclusively on fishes.

Text was added in order to clarify this issue.

Lines 122-133. Percent occurrence of prey in stomachs determines trophic level? Can you offer further explanation of this?

Relative abundance of prey in stomachs is used in order to determine trophic level. “Based on the %N contribution of each prey, the fractional trophic level (TROPH) of the species was estimated, using the routine for qualitative data of TrophLab.”

Figure 1:

Include a North arrow

A North arrow is included.

What does the asterisks represent? Sampling site? It should be stated in the figure caption.

Asterisk represents the only sampling site. It is now stated in the figure caption and in the text.

Depth contours should also be stated in the figure caption.

Depth contours are stated now in the figure caption.

Table 1:

Caption: are these data a summary from all hauls/ sampling events? Include sample sizes of hauls/ capture events.

Hauls/sampling occasions are now included in the caption of Table 1 according to reviewer’s comment.

How is the table sorted? There doesn't appear to be a logical order. Sort it alphabetically by family or number of individuals high to low.

Species were in alphabetical order. According to reviewer’s comment, species are now sorted alphabetically by family.

Is the name of the individual that first described the species necessary to include? If so, be sure to put all in parentheses.

In taxonomic studies, the name of the individual that first described the species is necessary to be included. Nevertheless, if the name of the species has subsequently been changed, then the name of the first recorder and the date are written in parentheses.

Why is there a size range for some fish and not for others? Are some averages of all individuals captured or subsamples?

Size range of TL is included for all fish species except for Lagocephalus sceleratus as only one individual was caught and examined. Size range of Lm is given only for Gobius niger, Parablennius tentacularis, Sphyraena sphyraena and Stephanolepis diaspros due to different references from different areas of the Mediterranean Sea. For all the other fish species only one value was given according to Fishbase. Table includes all individuals examined in the present study for stomach content analysis.

Results:

Line 154. Delete "Overall"

It is deleted.

Lines 172- 181. Run-on sentence, please break up!

Sentence was broken up.

Line 227. Insert 'and' between Caprella spp. and A. spinicornis

It was deleted after Reviewer’s 1 comment to shorten Results section.

Line 236. Please give some numbers, relative percentages, something, so the reader does not have to constantly go back to Appendix A and B.

Relative percentages are given.

Lines 237-238. Again, include numbers. And should there be a reference to Appendix A?

Numbers are included. Reference to Appendices A and B.

Line 241. You assume accidental prey, but there is no discussion on why that should be assumed in the discussion or a citation given in the results to back that up.

Reference and relevant text was added in the Discussion section, third sentence before the end of the first paragraph.

Lines 239- 254. A mix of results and discussion. Separate appropriately.

Results and discussion were separated appropriately. Text was transfered to the Discussion section.

Line 250. Missing a parentheses.

Parentheses added.

Discussion:

Lines 268 and 269. Commas missing

Commas added.

Line 282. Delete "and references therein"

It is deleted.

Lines 286-288. Awkward, rewrite.

Text was re-written in order to clarify this issue.

Lines 274-291. Smaller sized fish feeding on smaller prey? Unclear

Results of the present study have shown that zooplanktivorous fish species (small or larger individuals) examined in the study area preferred planktonic copepods which are small but also abundant in the marine environment. However, juveniles of many of the fish species examined which are in general characterized by benthic trophic behavior, were feeding on the small and abundant planktonic copepods. Text was corrected in order to clarify this issue. Also a sentence with a reference on this issue was added.

Line 313. What do you mean by "optimum feeding conditions"? Please describe.

Text was modified in order to clarify this issue.

In general, rewrite the discussion so there is less repetition of species by species results and more interpretation of those results-- what does it actually mean? eg lines 322-335 was good discussion material.

The discussion was re-written in order to have less results and more interpretation.

I would have liked to see a lot more discussion on the application of this knowledge. 1) Which of these fish species are commercially or recreationally fished? 2) Which are important prey species to predators in the area? 3) Or provide key food for commercial fish species? 4) What are the current threats to this system and how will this information help mitigate those threats? 5) How unique is this habitat? 6) Can these fish species be used as a proxy for other areas? 7) Were there any surprises from this study?

(1) Fish species that are commercially fished are now reported. Text was added in order to clarify this issue in the discussion section. (2) The semi-enclosed marine area of Elounda is characterized by the absence of fish predators, except for a few individuals of Sphyraena sphyraena which were juveniles, and this is one of the reasons why this particular marine area is a nursery ground and a refuge from predation (question 5, too). Text is added in the second sentence before the end of the first paragraph of the Discussion section. (3) Key food for commercial pelagic fish species (e.g. Spicara smaris, Spicara maena, Atherina boyeri, Boops boops) are planktonic copepods. But also for juveniles of commercial demersal fish species (e.g. Mullus barbartus, M. surmuletus, P. acarne) in the study area are planktonic copepods. Apart from copepods, small crustaceans are the key food which is explained in the second paragraph before the end of the discussion text. Text was added and modified in order to clarify these issues throughout the Discussion text. Moreover and concerning questions on fishes, study for fish assemblages in the study area is already published (Koulouri et al., 2016). The present study investigates the trophic structure of these fish species, which comprises basic knowledge essential for further monitoring of the area and holistic management appraches. (4) Tourist activities are the major threat for the study area as large boats are transferring tourists to and from Spinalonga island on a daily basis during the summer period (see Fig. 1) causing, by using high speed, resuspension of the sediments that could have a negative effect on the conservation of the C. prolifera bed and therefore on maintenance of fish and invertebrate assemblages as well as their juveniles. Flood events also take place in the study area occasionally. (6,7) Information given in the present and in previous studies in the study area concerns basic knowledge of this particular marine ecosystem with invertebrate and fish assemblages which are of similar diversity and therefore can be used as a proxy in other areas covered either by seagrass or macroalgal meadows also comprising nursery grounds in the Mediterranean Sea. Text was added in the last paragraph of Discussion in order to clarify all these issues.

Round 2

Reviewer 2 Report

Overall, a significant improvement over the previous version with the suggested modifications incorporated. 

Some fairly minor suggestions:

Abstract:

Line 20. "..characterised by high organic input and diversity of macrobenthic community..." Do you mean a highly diverse macrobenthic community?

Move the sentence beginning with "Feeding patterns of the fish, investigated on the basis of stomach content..." to before the sentence "1,642 fish, belonging to 17..."

 Line 27-28: insert "relative availability of" before 'macrobenthic'

Line 30: "...Fundamental for their survival in this particular habitat." Why?

Last sentence is very vague-- rewrite. 

Introduction:

Line 48: insert comma after 'structure'

Line 58: "are thought of having"-- change

Line 62: Change "Especially" to "In particular,"

Line 65. Is this paper studying the presence and effect of invasive species? In not, re-write to state that trophic knowledge is important to track the impacts of invasive species. 

Line 71: Insert a comma after 'seagrass meadows'

Last paragraph of introduction: much improved!

Methods:

2.2. sampling design

Still a bit unclear. I suggest deleting 'bimonthly', because that isn't accurate unless you did 12 hauls, not 6. End the first sentence after 'daylight'. The next sentence would be "Each haul swept an area approximately 0.006 km2." Then, "A single haul was taken in each of the following months: May 2006, July 2006..... for a total of six sampling hauls at one sampling site (Figure 1)."

Line 175: Typo. I believe you mean November 2006

Results:

Line 218: delete "on an occasional basis"

Discussion: 

Line 309: change 'Seasonal' to 'Temporal' since you did not in fact compare seasons.

Line 342: Change the sentence beginning with "Indeed, as it has been shown" to something like "This corresponds to a study in XX place where fishes in their early life history tend to a planktonic-oriented diet, regardless of their feeding habits as adults (Karachle and Stergiou 2013).

Line 389: delete 'also comprising' and change to 'which are also'

Line 393: delete 'by using high speed'

Author Response

Response to Review report (Round 2)

Overall, a significant improvement over the previous version with the suggested modifications incorporated.

Some fairly minor suggestions:

Abstract:

Line 20. "..characterised by high organic input and diversity of macrobenthic community..." Do you mean a highly diverse macrobenthic community?

Yes. Text is corrected.

Move the sentence beginning with "Feeding patterns of the fish, investigated on the basis of stomach content..." to before the sentence "1,642 fish, belonging to 17..."

The sentence is moved.

 Line 27-28: insert "relative availability of" before 'macrobenthic'

It is inserted.

Line 30: "...Fundamental for their survival in this particular habitat." Why?

Text is modified in order to clarify this issue.

Last sentence is very vague-- rewrite.

It is re-written.

Introduction:

Line 48: insert comma after 'structure'

Comma is inserted.

Line 58: "are thought of having"-- change

It is changed.

Line 62: Change "Especially" to "In particular,"

It is changed.

Line 65. Is this paper studying the presence and effect of invasive species? In not, re-write to state that trophic knowledge is important to track the impacts of invasive species.

Our intention is to emphasize that, studies on trophic structure of fish assemblages in marine macro-algal-dominated habitats concern only those of non-indigenous macroalgal species invaded in Mediterranean habitats. Text is modified in order to clarify this issue.

Line 71: Insert a comma after 'seagrass meadows'

It is inserted

Last paragraph of introduction: much improved!

Methods:

2.2. sampling design

Still a bit unclear. I suggest deleting 'bimonthly', because that isn't accurate unless you did 12 hauls, not 6. End the first sentence after 'daylight'. The next sentence would be "Each haul swept an area approximately 0.006 km2." Then, "A single haul was taken in each of the following months: May 2006, July 2006..... for a total of six sampling hauls at one sampling site (Figure 1)."

Text is corrected.

Line 175: Typo. I believe you mean November 2006

Text is corrected.

Results:

Line 218: delete "on an occasional basis"

It is deleted.

Discussion:

Line 309: change 'Seasonal' to 'Temporal' since you did not in fact compare seasons.

It is changed.

Line 342: Change the sentence beginning with "Indeed, as it has been shown" to something like "This corresponds to a study in XX place where fishes in their early life history tend to a planktonic-oriented diet, regardless of their feeding habits as adults (Karachle and Stergiou 2013).

Text is corrected.

Line 389: delete 'also comprising' and change to 'which are also'

Text is corrected.

Line 393: delete 'by using high speed'

It is deleted.